# SPiDR: A Simple Approach for Zero-Shot Safety in Sim-to-Real Transfer

**Yarden As**[*]
ETH Zurich

**Chengrui Qu**
Caltech

**Benjamin Unger**
ETH Zurich

**Dongho Kang**
ETH Zurich

**Max van der Hart**
ETH Zurich

**Laixi Shi**
Johns Hopkins University

**Stelian Coros**
ETH Zurich

**Adam Wierman**
Caltech

**Andreas Krause**
ETH Zurich

## Abstract

Deploying reinforcement learning (RL) safely in the real world is challenging, as policies trained in simulators must face the inevitable 'sim-to-real gap'. Robust safe RL techniques are provably safe however difficult to scale, while domain randomization is more practical yet prone to unsafe behaviors. We address this gap by proposing SPiDR, short for **S**im-to-real via **P**ess**i**mistic **D**omain **R**andomization—a scalable algorithm with provable guarantees for safe sim-to-real transfer. SPiDR uses domain randomization to incorporate the uncertainty about the sim-to-real gap into the safety constraints, making it versatile and highly compatible with existing training pipelines. Through extensive experiments on sim-to-sim benchmarks and two distinct real-world robotic platforms, we demonstrate that SPiDR effectively ensures safety despite the sim-to-real gap while maintaining strong performance.

## 1 Introduction

Reinforcement learning (RL) has made significant strides in recent years, demonstrating remarkable progress across a range of domains. These include achieving superhuman capabilities in games (Mnih et al., 2015; Silver et al., 2016), fine-tuning large language models (Ouyang et al., 2022), advancing applications in healthcare (Fox et al., 2020; Zhu et al., 2020), robotics (Lee et al., 2020; Degrave et al., 2022; Lin et al., 2025) and autonomous driving (Cusumano-Towner et al., 2025; Cornelisse et al., 2025). Yet despite these achievements, ensuring safety and preventing harmful behaviors remains a critical challenge and a prerequisite for unlocking the full potential of RL as a ubiquitous element in everyday life (Amodei et al., 2016; Gu et al., 2022).

The use of simulators has been a key component behind the success of many of the mentioned applications (Visentin et al., 2014; Makoviychuk et al., 2021; Degrave et al., 2022; Kazemkhani et al., 2024). Training in simulation allows agents to learn from unsafe interactions, which in reality would lead to catastrophic outcomes. In addition, learning complex behaviors fully online can be prohibitively time-consuming. Modern simulators accelerate training, reducing hours of real-world experience to minutes on consumer-grade GPUs (Rudin et al., 2022). However, while being a major driver in the development of the above examples, even state-of-the-art simulators often fall short in precisely mirroring the real-world. Indeed, "all models are wrong" (Box, 1976)—the so-called *sim-to-real gap* can make simulation-trained policies violate real-world constraints, which can be particularly dangerous in high-stakes settings where safety must be guaranteed on first contact.

Existing literature to address this challenge often relies on tools from robust optimization (Queeney and Benosman, 2024; Kitamura et al., 2024; Zhang et al., 2024). While being theoretically grounded, such methods typically require practitioners to significantly alter their existing training pipelines,

---

[*]Corresponding author: `yardas@ethz.ch`

39th Conference on Neural Information Processing Systems (NeurIPS 2025).

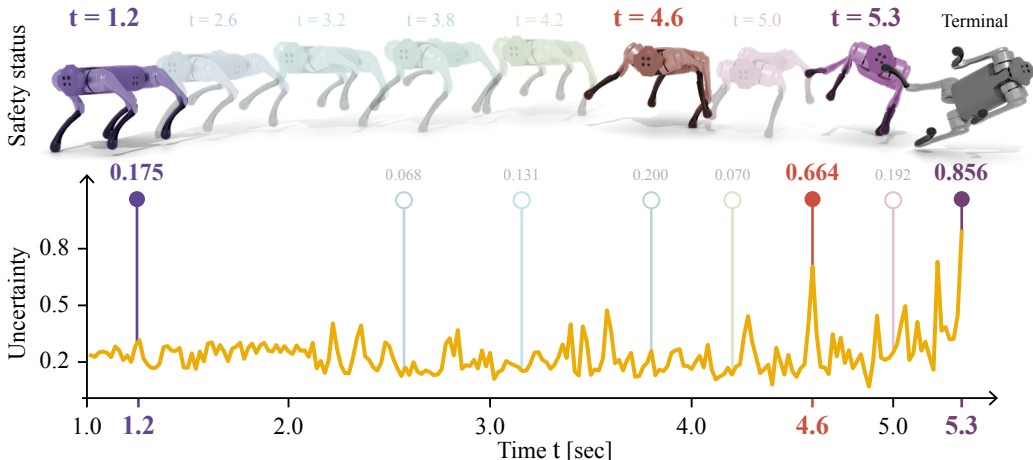

Figure 1: Uncertainty over a quadruped robot's trajectory. The snapshots illustrate the robot's pose at key moments, with corresponding uncertainty levels highlighted. High-uncertainty transitions are incorporated into the cost function to discourage the policy from entering regions where the simulator is inaccurate and behavior is more likely to become unsafe during real-world deployment.

rendering them less prevalent in practice. In contrast, due to its simplicity, domain randomization has become the de facto tool for sim-to-real transfer (Tobin et al., 2017; Peng et al., 2018; Lee et al., 2020; Degrave et al., 2022). Despite its success, in problems that require adherence to safety constraints, domain randomization lacks safety guarantees and often fails to satisfy the constraints in practice (cf. Queeney and Benosman, 2024, and Figure 3). Therefore, a method that provably guarantees safe sim-to-real transfer, while being highly compatible with standard training practices, is still missing.

In this work, we address this gap by presenting a simple method that builds on domain randomization while ensuring safety under sim-to-real transfer. We theoretically show that unsafe transfer can be associated with large *uncertainty* about the sim-to-real gap, quantified as the disagreement among next-state predictions from domain-randomized dynamics models. This key idea is illustrated in Figure 1, where spikes in uncertainty (e.g. at $t = 4.6$ and $t = 5.3$) coincide with unstable or unsafe behaviors, such as stumbling or flipping. Motivated by this insight, we propose to penalize the cost with the uncertainty to achieve safe sim-to-real transfer, leading to the design of SPiDR. Notably, SPiDR seamlessly integrates with state-of-the-art RL algorithms (Schulman et al., 2017; Haarnoja et al., 2019), delivering strong empirical performance on both in simulated and real-world safe RL tasks while ensuring constraint satisfaction, even under severe model mismatch.

**Out contribution.**

- We address an important challenge to real-world adoption of RL: zero-shot safe sim-to-real transfer, where an agent must learn a safe and effective policy using only simulated interactions. We propose SPiDR, a practical algorithm with formal safety guarantees that integrates easily into popular sim-to-real pipelines.

- We validate SPiDR on two real-world robotic platforms, where it achieves zero-shot constraint satisfaction, substantially outperforming other baselines in terms of safety and performance. These results provide empirical evidence that our theoretical guarantees translate to the real-world, suggesting that SPiDR can be safely used in real-world deployment.

- Finally, we extensively evaluate SPiDR on well established simulated continuous control benchmarks, including the RWRL benchmark (Dulac-Arnold et al., 2020), Safety Gym (Ray et al., 2019) and RaceCar environments (Kabzan et al., 2020), where SPiDR consistently satisfies safety constraints while achieving strong task performance.

## 2 Related Works

Safe sim-to-real transfer can be naturally framed as a constrained Markov decision process (CMDP, Altman, 1999) under model uncertainty. A common approach is to extend CMDPs using tools from robust optimization (Iyengar, 2005; Ben-Tal et al., 2009), which has led to a growing body

of work spanning both theoretical and practical contributions. We refer to Appendix N for a more comprehensive discussion of each line of work.

**Provably robust algorithms.** Zhang et al. (2024) build on a game-theoretic formulation to develop a tractable primal-dual algorithm with provable non-asymptotic convergence guarantees to a safe policy. Kitamura et al. (2024) take this further by proposing a policy gradient algorithm with formal guarantees for safety *and* optimality, via an epigraph form of the robust CMDP problem. Robust CMDPs are inherently challenging since the worst-case scenarios w.r.t. the reward and cost can differ (see Kitamura et al., 2024); both works make notable theoretical progress on this front. While sharing the goal of provably safe transfer, our work adopts a more scalable and modular approach by building on domain randomization, and integrating with existing CMDP solvers, avoiding the complexity of solving the minimax formulation, common in robust optimization.

**Scalable robust algorithms.** Russel et al. (2020) and Mankowitz et al. (2020) were among the first to study robust CMDPs in the context of deep RL, proposing practical methods that scale to continuous control tasks. Similarly, Queeney and Benosman (2024) introduce RAMU, an algorithm that uses coherent risk measures (Shapiro, 2017) with temporal differences (TD) learning, achieving strong empirical performance on the RWRL benchmark, though lacking formal safety guarantees. Sun et al. (2024) extends CPO (Achiam et al., 2017) to problems with model uncertainty, providing safety and performance guarantees. Lastly, Bossens (2024) proposes to learn an adversary policy and show that their algorithm is robust w.r.t. $L_1$-norm uncertainty sets. Our work differs in that it provides safety guarantees while remaining scalable and not tied to a particular RL algorithm.

**Practical methods for safe sim-to-real.** While the previous works develop methods for solving robust CMDPs—often motivated by the practical problem of safe sim-to-real transfer—other prior works address this problem directly. Kaushik et al. (2022) use online data from the real system and employ safe Bayesian optimization (Sui et al., 2015) to select safe and high-performing policies from a collection of policies trained in simulation. Similarly, Hsu et al. (2023) propose a multi-fidelity approach (Cutler et al., 2014), incorporating a fine-tuning step in a high-fidelity simulator prior to deployment. Both works emphasize practical applicability and provide strong empirical validation in real robotics settings. Compared to these works, in this work we focus on guaranteeing safe transfer without access to online data or a computationally expensive intermediate simulator.

## 3 Problem Setting

**Constrained Markov decision process.** We study discounted infinite-horizon CMDPs, defined by a tuple $\mathcal{M} = (\mathcal{S}, \mathcal{A}, p, r, c, \gamma, \rho)$. Here, $\mathcal{S}$ and $\mathcal{A}$ are the state and action spaces, $p : \mathcal{S} \times \mathcal{A} \to \Delta(\mathcal{S})$ denotes the transition probability of the system dynamics, the reward function is given by $r : \mathcal{S} \times \mathcal{A} \to [0, r_{\max}]$ and the cost is given by $c : \mathcal{S} \times \mathcal{A} \to [0, c_{\max}]$. The discount factor is $\gamma \in [0, 1)$ and $\rho \in \Delta(\mathcal{S})$ is a probability distribution from which initial states are drawn. We consider the class of stationary policies $\Pi^s$, where each policy is a stochastic mapping from states to actions $\pi : \mathcal{S} \to \Delta(\mathcal{A})$.[2] For given dynamics $p$, the value function under policy $\pi$ at state $s$ is defined as $V_r^{p,\pi}(s) \triangleq \mathbb{E}_{p,\pi}[\sum_{t=0}^{\infty} \gamma^t r_t \mid s]$. Similarly, the cost value is given by $V_c^{p,\pi}(s) \triangleq \mathbb{E}_{p,\pi}[\sum_{t=0}^{\infty} \gamma^t c_t \mid s]$. We define the expected value function of $\pi$ when the initial state is sampled from $\rho$ as $J_p(\pi) \triangleq \mathbb{E}_{s \sim \rho}[V_r^{p,\pi}(s)]$, and the expected cost value as $C_p(\pi) \triangleq \mathbb{E}_{s \sim \rho}[V_c^{p,\pi}(s)]$. The goal is to find a policy $\pi$ that solves

$$\max_{\pi \in \Pi^s} J_p(\pi) \quad \text{s.t.} \quad C_p(\pi) \leq d, \tag{1}$$

where $d > 0$ is a predefined budget. This formulation enables explicit decoupling of safety from the objective. For example, in robotics, the cost can represent collisions with obstacles, while the reward encourages reaching a goal.

**Safe sim-to-real transfer.** In this work, we consider a setting where the agent has access to a simulator, capable of generating any environment $\widehat{\mathcal{M}}_\xi = (\mathcal{S}, \mathcal{A}, \hat{p}_\xi, r, c, \gamma, \rho)$ given $\xi \in \Xi \subseteq \mathbb{R}^{d_\xi}$. The dynamics of each environment are parameterized by $\xi$, typically representing the physical properties of the system dynamics. The agent can freely interact with any simulated environment $\hat{p}_\xi$ with $\xi \in \Xi$, but has no access to the unknown real environment $\mathcal{M}^\star = (\mathcal{S}, \mathcal{A}, p^\star, r, c, \gamma, \rho)$. The

---

[2]While only the class of history-dependent policies is formally complete under domain randomization, we focus on $\Pi^s$ due to its simplicity. Our theoretical and empirical results can be directly extended to history-dependent policies. See Dolgov and Durfee (2005); Kwon et al. (2021); Chen et al. (2022) for further discussions.

objective is to learn a policy $\pi \in \Pi^s$ purely within the simulator such that, when deployed in the real environment $\mathcal{M}^\star$, it satisfies the safety constraint $C_{p^\star}(\pi) \leq d$. Crucially, the agent must guarantee constraint satisfaction "zero-shot", without any direct interaction with the real environment $\mathcal{M}^\star$.

# 4  SPiDR for Safe Zero-Shot Sim-to-Real Transfer

## 4.1  Domain Randomization

**Extending domain randomization to CMDPs.**  Domain randomization is particularly well-suited for the problem setting described above, as it leverages a set $\{\xi_i\}_{i=1}^N \overset{\text{i.i.d}}{\sim} \mu$ of parameterized environments, sampled independently from some probability distribution $\mu$. This distribution acts as a *prior* for the real, yet unknown system parameters. A natural approach for tackling safe sim-to-real problems is by formulating CMDPs over a distribution of randomized domains, i.e., solving

$$\max_{\pi \in \Pi^s} \mathbb{E}_{\xi \sim \mu} J_{\hat{p}_\xi}(\pi) \quad \text{s.t.} \quad \mathbb{E}_{\xi \sim \mu} C_{\hat{p}_\xi}(\pi) \leq d. \tag{2}$$

Domain randomization can be seen as a sample average approximation of Equation (2), making it both straightforward to implement and scalable, as massively-parallel simulators can be used to collect data from each environment in parallel.

**Domain randomization is not always safe.**  While Equation (2) provides a compelling formulation from a practical standpoint, it does not guarantee safety in the real environment $\mathcal{M}^\star$. Specifically, since simulators only approximate the real world with limited precision, as well as due to averaging over dynamics, the costs in Equation (2) may *underestimate* the true costs in $\mathcal{M}^\star$. This limitation is empirically validated in Sections 5.1 and 5.2 (Figures 3 and 6, respectively) and theoretically illustrated through an example in Appendix A. In what follows, we formally characterize constraint underestimation and show how SPiDR is designed to mitigate it.

**A pessimistic upper bound.**  We quantify the extent by which constraints on the real system may be underestimated by establishing the following bound. To this end, we measure the discrepancy between the simulated and real dynamics using the $L_1$-Wasserstein distance, denoted as $D_W(\hat{p}_\xi, p^\star)(s, a)$, whose formal definition is provided in Definition C.1. While our analysis is based on this metric, it naturally extends to other discrepancy measures. We assume this discrepancy is finite for all $\xi \in \Xi$, which is a reasonable assumption in practice. For instance, the simulators by Makoviychuk et al. (2021) and Zakka et al. (2025) have been successfully used for zero-shot sim-to-real transfer across several robotic platforms, suggesting that they maintain high fidelity with real-world systems. We now present our bound below.

**Lemma 4.1.** *Let $\mathbb{P}_{p,\pi,t}(s)$ denote the probability of reaching the state $s$ at step $t$ under the policy $\pi$ and the dynamics $p$, and let $d_{p,\pi} \triangleq (1 - \gamma)\pi(a|s)\sum_{t=0}^\infty \gamma^t \mathbb{P}_{p,\pi,t}(s)$ denote the normalized discounted occupancy measure of policy $\pi$ under the dynamics $p$. The real-world cost $C_{p^\star}(\pi)$ can be upper-bounded by*

$$C_{p^\star}(\pi) \leq \underbrace{\mathbb{E}_{\xi \sim \mu} C_{\hat{p}_\xi}(\pi)}_{\textit{Constraint in simulation}} + \mathbb{E}_{\xi \sim \mu}\left[\mathbb{E}_{(s,a)\sim d_{\hat{p}_\xi,\pi}}\left[\frac{\gamma L_C}{1-\gamma} D_W(\hat{p}_\xi, p^\star)(s, a)\right]\right], \tag{3}$$

*where $L_C$ is the Lipschitz constant of the state cost function $V_c^{p^\star, \pi}(s)$.*

Lemma 4.1 shows that the true safety constraint function $C_{p^\star}(\pi)$ is upper-bounded by the constraint evaluated during training, and the expected $L_1$-Wasserstein distance with respect to the state-action occupancy measure of $\pi$ over the simulated dynamics $\hat{p}_\xi$. Importantly, it establishes that even when the constraint is satisfied in simulation, i.e., $\mathbb{E}_{\xi \sim \mu} C_{\hat{p}_\xi}(\pi) \leq d$, the constraint on the real system may be larger, depending on $\pi$ and the degree of mismatch between the simulated and real dynamics. Therefore, by bounding the r.h.s. of Equation (3) with $d$, we guarantee that $C_{p^\star}(\pi) \leq d$, hence safe transfer to the real system. We refer to Appendix B for the formal proof and assumptions of Lemma 4.1. We next show how this key insight is used in our design of SPiDR.

## 4.2  Algorithm Design

**Reduction to penalized CMDPs.**  Observing that by linearity of expectation, the r.h.s. of Equation (3) can be written as

$$\mathbb{E}_{\xi \sim \mu}\left[\mathbb{E}_{(s,a)\sim d_{\hat{p}_\xi,\pi}}\left[c(s, a) + \frac{\gamma L_C}{1-\gamma} D_W(\hat{p}_\xi, p^\star)(s, a)\right]\right].$$

This simple insight motivates the use of

$$\tilde{c}(s, a) \triangleq c(s, a) + \underbrace{\frac{\gamma L_C}{1 - \gamma} \max_{\xi \in \Xi} D_W(\hat{p}_\xi, p^\star)(s, a)}_{\text{penalty}} \quad (4)$$

as a surrogate cost function during training. Crucially, it is infeasible to estimate the model discrepancy $D_W(\hat{p}_\xi, p^\star)(s, a)$ for any $\xi \in \Xi$, therefore we use the worst-case one as a conservative approximation. Using $\tilde{c}(\cdot, \cdot)$ yields a *penalized* CMDP $(\mathcal{S}, \mathcal{A}, \hat{p}_\xi, r, \tilde{c}, \gamma, \rho)$. This CMDP is still fully compatible with domain randomization, allowing us to solve

$$\max_{\pi \in \Pi^s} \mathbb{E}_{\xi \sim \mu} J_{\hat{p}_\xi}(\pi) \quad \text{s.t.} \quad \mathbb{E}_{\xi \sim \mu} \widetilde{C}_{\hat{p}_\xi}(\pi) \leq d, \quad (5)$$

where $\widetilde{C}_{\hat{p}_\xi}(\pi)$ denotes the constraint with $\tilde{c}(\cdot, \cdot)$ following $\hat{p}_\xi$. While $\tilde{c}(\cdot, \cdot)$ can be used as a conservative approximation that in principle guarantees safe transfer, direct access to the penalty term is generally intractable. This is due to the fact that $L_C$ and $D_W(\cdot, \cdot)$ are unknown a priori and can only be estimated using access to ground-truth data from the real system. Therefore, we propose approximating the penalty term using only simulated data.

**Approximating the penalty term.** We propose using an ensemble of $\{\hat{p}_{\xi_i}\}_{i=1}^n \overset{\text{i.i.d}}{\sim} \mu$ dynamics and measuring their disagreement in predicting the next state, as a proxy for the uncertainty about the model discrepancy $\max_{\xi \in \Xi} D_W(\hat{p}_\xi, p^\star)$. Specifically, when $s_i \sim \hat{p}_{\xi_i}(\cdot \mid s, a)$ are $n$ i.i.d. samples, we define the sum of component-wise empirical variances of the ensemble next-state predictions as the estimator

$$v(s, a) \triangleq \|\text{Var}(s_1, \ldots, s_n)\|_1 = \sum_{j=1}^{\dim(\mathcal{S})} \text{Var}(s_{1,j}, \ldots, s_{n,j}) \quad (6)$$

where $\dim(\mathcal{S})$ is the dimension of the state space and $s_{i,j}$ denotes the $j$-th component of the $i$-th sample. The empirical variance $v(s, a)$ measures the sensitivity of the environment in predicting the next state w.r.t. $\xi$, making it an effective proxy for our uncertainty about the model discrepancy $\max_{\xi \in \Xi} D_W(\hat{p}_\xi, p^\star)$, especially when the real environment lies near the simulated ones. In Appendix C, we formally show that for a suitable constant $\lambda$, and under bounded model mismatch, $\lambda v(s, a)$ upper-bounds $\frac{\gamma L_c}{1-\gamma} \max_{\xi \in \Xi} D_W(\hat{p}_\xi, p^\star)(s, a)$, with the bound becoming tighter as $n$ increases. Additional practical guidance on how to pick $\lambda$ empirically is provided in Appendix E. We find this approach simple to implement, computationally efficient and effective in practice, as demonstrated in Section 5 (Figures 3 and 7). Moreover, in Figure 8 we demonstrate how to pick $n$ in practice.

**The algorithm.** With the above approximation $v(\cdot, \cdot)$ in hand, we are ready to introduce the entire algorithm SPiDR, summarized in Algorithm 1. Standard domain randomization typically involves policy search methods such as policy gradients or TD learning, which require collecting trajectory data from the simulator. These trajectories are collected independently and in parallel for each dynamics $\{\xi_i\}_{i=1}^N$. To incorporate pessimism, we modify only the procedure by which these trajectories are obtained. This abstraction allows us to keep using domain randomization as it is while remaining versatile w.r.t. the choice of policy search algorithm. Specifically, following standard domain randomization, SPiDR samples a batch of dynamics $\{\xi_i\}_{i=1}^N$. These dynamics are used only to rollout the policy (Lines 2 and 7). For each sampled dynamics, we further employ an ensemble $\{\hat{p}_{\xi_{ij}}\}_{j=1}^n$. This ensemble is used to estimate the penalty term as $v(\cdot, \cdot)$ (Lines 8 to 10). These trajectories in the "penalized" CMDP using $\tilde{c}$ are collected and used to solve the penalized CMDP in Equation (5). Importantly, Lines 8 to 10 represent the only modifications made relative to standard domain randomization.

### 4.3 Safety Guarantee

Next, we demonstrate our theoretical guarantees for a solution to Equation (5). We first assume the feasibility of the problem, otherwise, a solution can be recovered only by improving the simulator or with safe online learning techniques (As et al., 2025). We show that any solution to the penalized CMDP provably achieves safety in the real environment by the following theorem.

**Theorem 4.2.** *Let $p^\star$ be the dynamics of the real environment $\mathcal{M}^\star$. Let $\tilde{\pi}$ be a solution to the penalized CMDP introduced in Equation (5), i.e., $\tilde{\pi} = \max_{\pi \in \Pi^s} \mathbb{E}_{\xi \sim \mu} J_{\hat{p}_\xi}(\pi)$ s.t. $\mathbb{E}_{\xi \sim \mu} \widetilde{C}_{\hat{p}_\xi}(\pi) \leq d$. Then $\tilde{\pi}$ satisfies the safety constraint in the real environment, namely, $C_{p^\star}(\tilde{\pi}) \leq d$.*

---

**Algorithm 1** SPiDR: Safe Sim-to-Real via Pessimistic Domain Randomization

---
1: **Input:** pessimism $\lambda$, initial distribution $\mu$, behavior policy $\pi$
2: **Init:** Sample $\{\xi_i\}_{i=1}^N \times \{\xi_{ij}\}_{j=1}^n \overset{\text{i.i.d}}{\sim} \mu$
3: **for** $i = 1, \dots, N$ **in parallel do**                                  ▷ Collect data from each $\xi_i$
4:     Initialize trajectory $\tau^{(i)} \leftarrow \emptyset$
5:     **for** $t = 0, 1, \dots$ **do**                                          ▷ Rollout policy
6:         $a_t \sim \pi(\cdot \mid s_t)$
7:         Simulate $s_{t+1} \sim \hat{p}_{\xi_i}(\cdot \mid s_t, a_t)$, obtain $r_t, c_t$
8:         Simulate $s_{t+1}^{(j)} \sim \hat{p}_{\xi_{ij}}(\cdot \mid s_t, a_t)$ with each $\{\hat{p}_{\xi_{ij}}\}_{j=1}^n$     ▷ Fast parallel execution
9:         Compute the penalty term $\upsilon(s_t, a_t)$ in Equation (6)
10:        Penalize cost $\tilde{c}_t \leftarrow c(s_t, a_t) + \lambda \upsilon(s_t, a_t)$
11:        Append $(s_t, a_t, r_t, \tilde{c}_t)$ to $\tau^{(i)}$
12:        Set next state $s_t \leftarrow s_{t+1}$
13:    **end for**
14: **end for**
15: Solve Equation (5) using $\{\tau^{(i)}\}_{i=1}^N$ to obtain $\tilde{\pi}$ with any CMDP solver
16: **return** $\tilde{\pi}$

---

The formal proof of this theorem, including its assumptions, is provided in Appendix B. Solving the CMDP with any penalty on the cost that dominates $\max_{\xi \in \Xi} D_W(\hat{p}_\xi, p^\star)$ yields a more conservative policy. Any such policy will remain safe in the real environment. The drop in performance due to cost penalization is not captured by Theorem 4.2 as it also depends on cost and reward function. However, our real-world experiments in Section 5 (Figure 3) demonstrate that SPiDR consistently matches or even outperforms standard domain randomization. Adopting any penalty on the cost reduces the set of feasible policies. Therefore, the agent may avoid high-reward regions that are in reality safe but uncertain in simulation. However, the limited drop observed in practice suggests that our bound $\upsilon(\cdot, \cdot)$ provides a reasonably tight over-approximation of the worst-case model discrepancy.

## 5 Experiments

Next, we demonstrate SPiDR's performance in practice. Below we provide our sim-to-real experiments on two robotic tasks, followed by comprehensive ablations on several tasks from three well-established safe RL benchmarks. We refer the reader to Appendices D and E for additional details on how $\lambda$ is picked in practice and for more ablations.

**Setup.** Unless otherwise specified, all experiments use SAC (Haarnoja et al., 2019; Nauman et al., 2024) in combination with either CRPO (Xu et al., 2021) or a simple primal-dual constrained optimization method (Bertsekas, 2016). We use these CMDP solvers since they deliver strong results while remaining easy to implement. We run each experiment with five random seeds and report the mean and standard error. Empirical estimates of the objective and constraint on the *test* environments are denoted by $\hat{J}(\tilde{\pi})$ and $\hat{C}(\tilde{\pi})$, respectively. We compare SPiDR with the following baselines: **(i)** Nominal is a simple baseline that collects trajectories only from the nominal training dynamics; **(ii)** Domain Randomization collects trajectories only from the perturbed dynamics of the training distribution; and **(iii)** RAMU (Queeney and Benosman, 2024), a state-of-the-art robust safe RL algorithm designed specifically for TD learning methods.

### 5.1 Real-World Deployment

We demonstrate SPiDR's effectiveness on two real-world robotic tasks: a highly-dynamic remote-controlled race car, and on a Unitree Go1 quadruped robot, illustrated in Figure 2. Each policy is trained only in simulation and then evaluated in the real world: five trials for the remote-controlled car and ten trials for Unitree Go1. These trials are used to obtain $\hat{C}(\tilde{\pi})$ and $\hat{J}(\tilde{\pi})$ where applicable. In both tasks, we compare SPiDR with domain randomization evaluated on the real system. Additionally, to demonstrate how domain randomization may underestimate the constraint in real, we compare its performance with SPiDR, when evaluated *during training in simulation*. Further details about the tasks and hardware are provided in Appendices G and H.

**Experiment 1: Does SPiDR transfer safely to real systems?** We present our results in Figure 3. As shown, in both tasks, SPiDR satisfies the constraint, whereas domain randomization dramatically

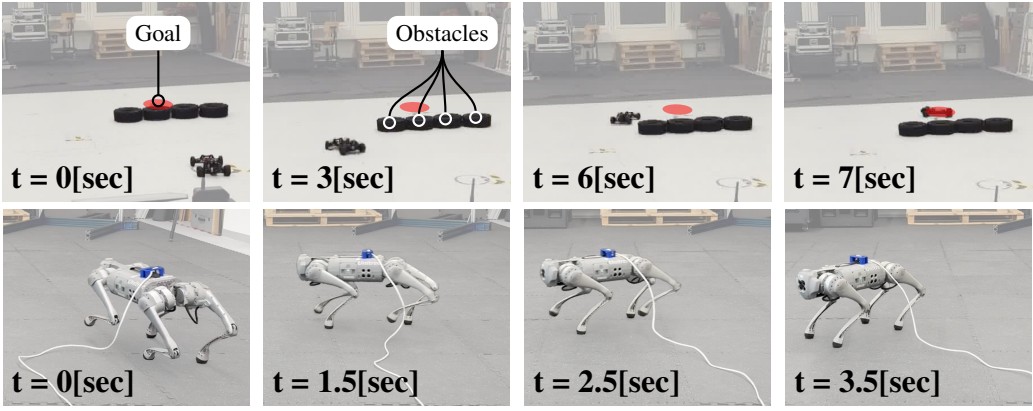

Figure 2: Example trajectories SPiDR with RaceCar and Unitree Go1.

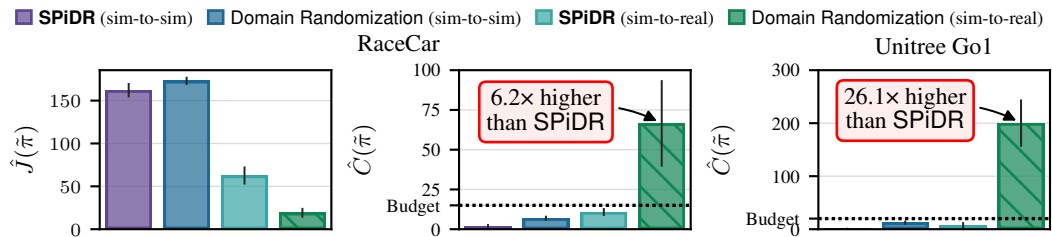

Figure 3: Performance on the race car and Unitree Go1. ■ SPiDR (sim-to-sim) and ■ SPiDR (sim-to-real) represent evaluation *in simulation* and on the *real system* respectively. SPiDR transfers safely, while domain randomization dramatically violates the safety constraints.

violates it, *even when the constraint is satisfied in simulation*. This result is consistent with Lemma 4.1, due to constraint underestimation in simulation. Remarkably, due to the highly dynamic behavior of the remote-control car, the performance of the objective with domain randomization underperforms compared to SPiDR. This is because domain randomization often overshoots the target, whereas SPiDR's more conservative hence slower policies avoid overshooting.

To evaluate the performance on the Unitree Go1 robot, we measure the rewards on the real system are report the performance in Figure 4. As demonstrated in Figure 4, both domain randomization and SPiDR exhibit comparable decrease in performance relative to their simulated counterparts. These results demonstrate that SPiDR transfers effectively to the real-world system without being overly conservative, successfully solving the task. We additionally provide five trajectories for each policy and algorithm, resulting in a total of 75 video demonstrations, provided in the following link. These recordings indicate that using SPiDR does not lead to a noticeable degradation in locomotion performance. In comparison, RAMU, trained with its default hyperparameters, succeeds in following the commands in $12 \pm 4.8\%$ of the trials, falling in the rest, while SPiDR completes all trials successfully without falling. See Appendix G for more details. These results suggest that SPiDR works well across different real-world robotic tasks, satisfying the constraints while maintaining strong performance.

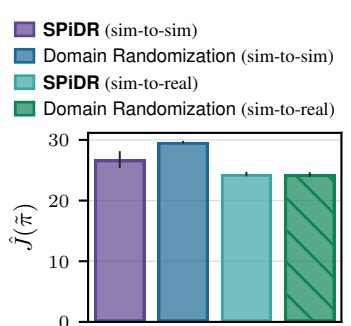

Figure 4: Performance is maintained on the Unitree Go1.

**Experiment 2: How versatile is** SPiDR? We replace SAC with PPO (Schulman et al., 2017) and a primal-dual optimizer as a CMDP solver, repeating our experiment on the Unitree Go1 robot. We present our results in Figure 5, demonstrating that SPiDR satisfies the constraints on the real system, while domain randomization with PPO satisfies the constraint *only in simulation*. We provide additional sim-to-sim experiments with PPO in Appendix I. Throughout this work, we use three different CMDP solvers, with two different policy search methods, demonstrating

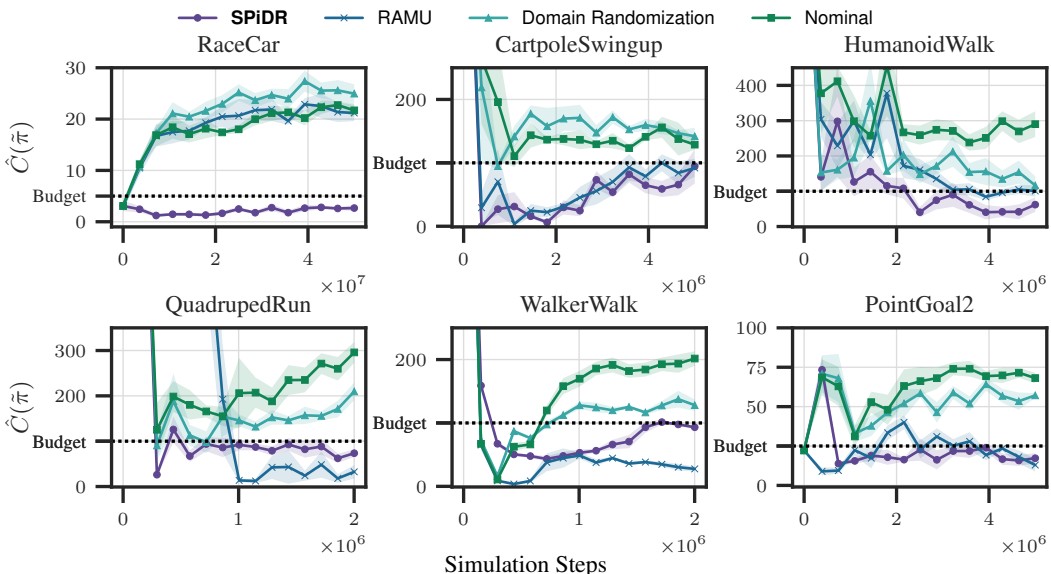

Figure 6: Costs over training iterations, evaluated on the simulated test environments. SPiDR consistently satisfies the constraints across all tasks.

competitive performance while remaining safe, including on real hardware. These results highlight the versatility and broad applicability of SPiDR across different CMDP solvers and RL algorithms.

## 5.2 Evaluation in the Sim-to-Sim Sandbox

Next, we demonstrate that SPiDR consistently achieves strong task performance while adhering to safety constraints in a series of sim-to-sim experiments. This set of experiments serves as a controlled testbed for systematically ablating SPiDR. We provide additional ablations on the robustness to choice of $\lambda$ empirical intuition about $\upsilon(\cdot, \cdot)$ in Appendix D. Full learning curves and further details on the experimental setup can be found in Appendices F and J to L.

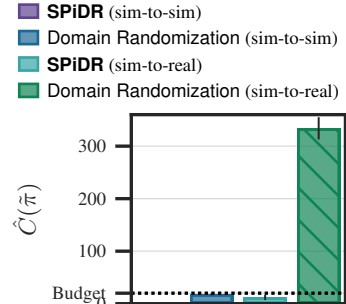

Figure 5: Safe transfer to a real Unitree Go1 with PPO.

**Experiment 3: Can SPiDR satisfy the constraints under sim-to-sim gap?** We demonstrate our results in Figure 6, reporting the learning curves of constraint across tasks. Figure 6 demonstrates that SPiDR satisfies the constraints on all six tasks. This is in contrast to Nominal and Domain Randomization, which fail to satisfy the constraints across all tasks, as expected. In the RaceCar task, where the sim-to-sim gap primarily arises due to the training simulator failing to capture all the relevant phenomena present in the test environment, SPiDR is the only algorithm that satisfies the constraint.

**Experiment 4: What is the tradeoff between safety and performance?** We study the tradeoff between constraint satisfaction and objective performance, showing that SPiDR does not suffer significant performance decrease. We present our results in Figure 7. As shown, SPiDR maintains balance of high performance while satisfying the constraints across all tasks. SPiDR and RAMU demonstrate comparable performance on the RWRL environments. On the other hand, on the RaceCar task, which involves more realistic and challenging model discrepancies, SPiDR significantly outperforms RAMU. Notably, in environments like HumanoidWalk, QuadrupedRun, WalkerWalk and RaceCar, SPiDR not only meets the safety constraints but also achieves competitive performance with respect to the optimal performance on these tasks. These results illustrate that performance is not significantly compromised, even in the presence of large model mismatch.

**Experiment 5: How does SPiDR scale?** We analyze how the performance of SPiDR is affected by the ensemble size $n$ in terms of training wall-clock time and performance. We test SPiDR on Walker-Walk, QuadrupedRun and HumanoidWalk while varying $n \in \{1, 2, 4, 8, 16, 32, 64, 128\}$. In Figure 8

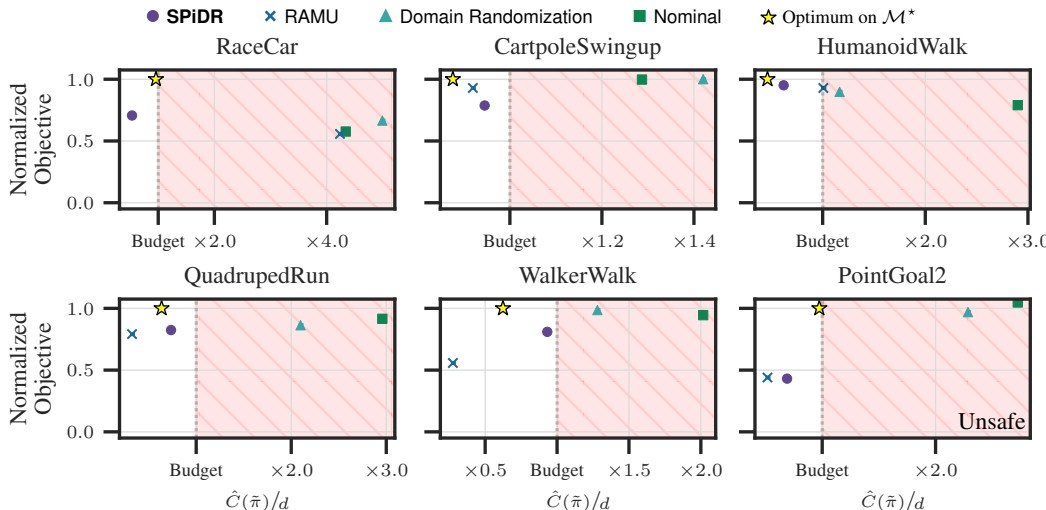

Figure 7: Normalized objective and constraint performance after training. The constraint is normalized relative to the budget, while the objective is normalized against the optimal performance on $\mathcal{M}^\star$. Shaded red area represents unsafe policies. SPiDR consistently satisfies the constraints while achieving competitive performance on different locomotion and navigation tasks.

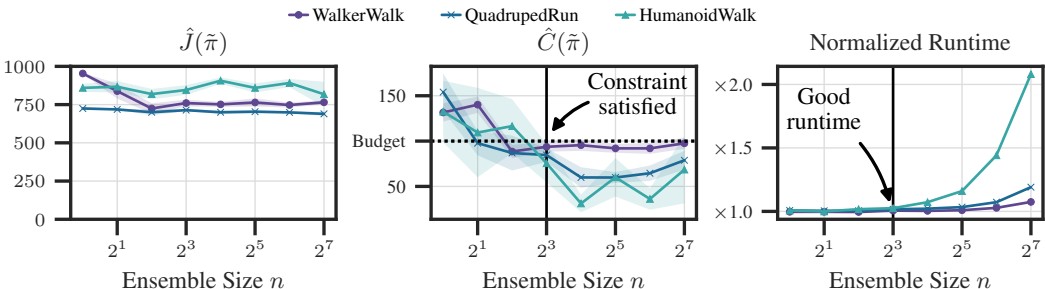

Figure 8: Runtime overhead of SPiDR is marginal even for $n \geq 8$. For $n \geq 8$ safety is maintained across environments.

we report the objective, constraint and the relative training runtime compared to standard domain randomization. As shown, for $n \geq 8$, the constraint remains within budget in all environments, while increasing $n$ has little effect on the objective. Furthermore, on all three environment the relative runtime for $n \leq 32$ is only marginally longer compared to standard training with domain randomization. All experiments are run using MuJoCo XLA (Freeman et al., 2021), enabling us to train SAC for 5M and PPO for 200M environment steps in under an hour on a single NVIDIA RTX 4090 GPU.

**Does SPiDR scale to vision control tasks?** We investigate whether SPiDR scales to partially observable vision-based settings, where the policy operates directly on rendered images. To evaluate this, we implement an asymmetric teacher–student Lee et al. (2020) setup: the policy receives only image observations, while the cost is penalized using privileged state information available in simulation, following Equation (6). Specifically, we combine DrQ (Yarats et al., 2021) with CRPO (Xu et al., 2021) as the penalizer and apply SPiDR on the CartpoleSwingup task. This setup follows our previous sim-to-sim experiment on the CartpoleSwingup task, only differing in the observation given to the agent. The policy trains on a sequence of 3 stacked grayscale 64×64 pixels images, while the cost penalty uses the true system states (not shown to the agent). We report our results on the evaluation environment in Figure 9. These results show that even in a partially observable setup, SPiDR satisfies the constraint by leveraging privileged information during training. Training the vision-based policy for 1M simulation steps with SPiDR takes roughly 33 minutes on an RTX 4090 GPU, compared to roughly 30 minutes without SPiDR. The main computational bottleneck lies in computing critic gradients, while simulation efficiency comes from Madrona-MJX (Rosenzweig et al., 2024), which renders rollouts from 128+ environments in parallel. This experiment suggests that SPiDR scales effectively to vision-based control tasks.

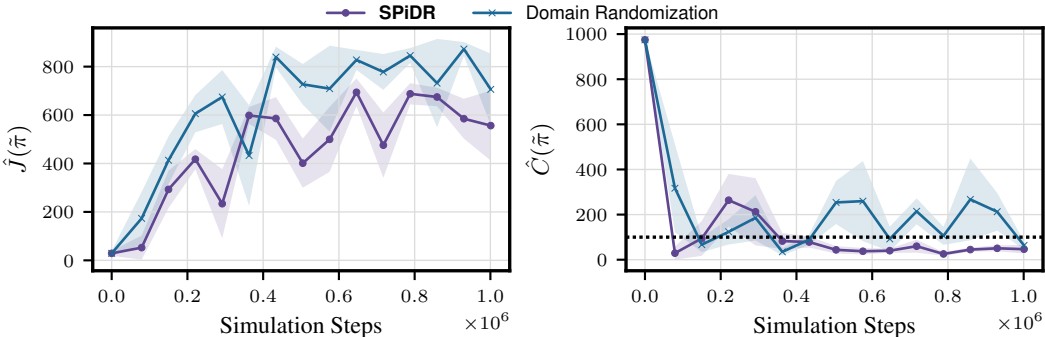

Figure 9: Learning curves of vision-based CartpoleSwingup task. SPiDR satisfies the constraints under dynamics mismatch even when the policy observes grayscale images of the cartpole.

## 6 Conclusions

In this work we address safe sim-to-real transfer, a key challenge limiting the broader adoption of RL in real-world applications. We theoretically link constraint violations upon deployment to uncertainty about the sim-to-real gap and propose a simple and provably safe algorithm that penalizes the cost function using an estimated gap under domain randomization. Our proposed method – SPiDR – is theoretically sound, easy to implement and can be readily combined with popular safe RL algorithms. We empirically show that SPiDR consistently achieves strong performance while maintaining safety across different tasks spanning three different simulated safe RL benchmarks. Moreover, SPiDR solves two real-world robotic tasks as it ensures safe transfer, proving its practical applicability to the full-scale problem it aims to address. Although we focus our experiments on robotic tasks, due to the simplicity of SPiDR, we believe that future work can extend it to other safety-critical domains. Finally, since our method focuses on the "zero-shot" transfer setting, where access to real-world data is restricted, proposing hybrid approaches that combine simulation-trained policies with safe online exploration techniques is an important direction for future work.

## Acknowledgments and Disclosure of Funding

We would like to thank Taerim Yoon, Yonathan Efroni, Kishan Panaganti, Chenhao Li and James Queeney for insightful discussions during the development of this project. We thank the anonymous reviewers for their valuable comments and suggestions. This project has received funding from a grant of the Hasler foundation (grant no. 21039) and the European Research Council (ERC) under the European Union's Horizon 2020 research and innovation programme (grant agreement No. 866480). Adam Wierman is supported by NSF through CNS-2146814, CPS-2136197, CNS-2106403, NGSDI2105648, IIS-2336236, and by the Resnick Sustainability Institute at Caltech.

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

# Appendix

## A   Failure modes of Domain Randomization

The following example illustrates how domain randomization can fail to ensure constraint satisfaction in practice, even if all the simulator environments are arbitrarily close to the real environment.

**Example A.1.** *Suppose $\mathcal{S} \geq 3$, $\mathcal{A} \geq 2$, then for all $0 < \varepsilon \leq 1/4$, there exists a set of simulated CMDPs $\{\mathcal{M}_i = (\mathcal{S}, \mathcal{A}, \hat{p}_{\xi_i}, r, c, \gamma, \rho)\}_{i=1}^{N}$, an unknown real CMDP $\mathcal{M}^{\star} = (\mathcal{S}, \mathcal{A}, p^{\star}, r, c, \gamma, \rho)$ satisfying*

$$D_W(p^{\star}, \hat{p}_{\xi_i})(s, a) \leq \varepsilon, \ \forall i \in \{1, \dots, N\}, (s, a) \in \mathcal{S} \times \mathcal{A}.$$

*with a fixed budget $d > 0$ for all CMDPs, such that the domain randomization policy $\pi_{DR}$ returned by Equation (2) with any $\mu$, will be unsafe in the real environment,*

$$C_{p^{\star}}(\pi_{DR}) > d.$$

*Proof.* Consider the CMDPs given in Figure 10, where there are three states, the initial state $s_0$, two absorbing states $s_1$ and $s_2$, two actions $a_1$ and $a_2$. The state space is in $\mathbb{R}$, and we assume $s_0 = 0$, $s_1 = 1$ and $s_2 = 2$. The reward and cost function are given by:

$$r(s_0, a) = 0, \ c(s_0, a) = 0, \ \forall a \in \mathcal{A};$$
$$r(s_1, a) = 1, \ c(s_1, a) = 1, \ \forall a \in \mathcal{A};$$
$$r(s_2, a) = 0, \ c(s_2, a) = 0, \ \forall a \in \mathcal{A};$$

In the $i$-th simulated environment, at the initial state $s_0$, the transition probability is given by:

$$\begin{cases} \hat{p}_{\xi_i}(s_0 \mid s_0, a) = 0, \ \forall a \in \mathcal{A}, \\ \hat{p}_{\xi_i}(s_1 \mid s_0, a) = \dfrac{1}{2} + \varepsilon \mathbb{1}\{a = a_1\}, \ \forall a \in \mathcal{A}, \\ \hat{p}_{\xi_i}(s_2 \mid s_0, a) = \dfrac{1}{2} - \varepsilon \mathbb{1}\{a = a_1\}), \ \forall a \in \mathcal{A}, \end{cases}$$

In the real environment, at the initial state $s_0$, the transition probability is given by:

$$\begin{cases} p^{\star}(s_0 \mid s_0, a) = 0, \ \forall a \in \mathcal{A}, \\ p^{\star}(s_1 \mid s_0, a) = \dfrac{1}{2} + \varepsilon(1 + \mathbb{1}\{a = a_1\}), \ \forall a \in \mathcal{A}, \\ p^{\star}(s_2 \mid s_0, a) = \dfrac{1}{2} - \varepsilon(1 + \mathbb{1}\{a = a_1\}), \ \forall a \in \mathcal{A}, \end{cases}$$

For all these environments, at the absorbing states $s_1$ and $s_2$, we have $p(s_2 \mid s_2, a) = p(s_1 \mid s_1, a) = 1$ for $\forall a \in \mathcal{A}$. Then the maximum Wasserstein distance can be bounded by epsilon, i.e. $D_W(p^{\star}, \hat{p}_{\xi_i})(s, a) \leq \varepsilon, \forall (s, a) \in \mathcal{S} \times \mathcal{A}$ for every simulated environment. Take the budget to be $d = \frac{\gamma}{1-\gamma}(\frac{1}{2} + \varepsilon)$.

To achieve the highest cumulative reward, the optimal domain randomization policies returned by Equation (2) is supported only on $a_1$ for any training distribution $\mu$, but the cost incurred by these policies in the real environment is $(\frac{1}{2} + 2\varepsilon)\frac{\gamma}{1-\gamma} > d$, violating the constraint in the real environment. $\square$

We see that a policy trained via standard domain randomization in the example Example A.1 can still be unsafe in the real environment. The only safe policy in the real environment is $\pi(s_0) = a_2$, which cannot be learned with pure domain randomization in this example. This unsafe transfer is not only due to the averaging cost constraint, but also due to inherent mismatch between any simulated and real dynamics. This mismatch may lead the learned policy to frequently visit regions with underestimated cost in simulation, but which incur high actual cost in the real world.

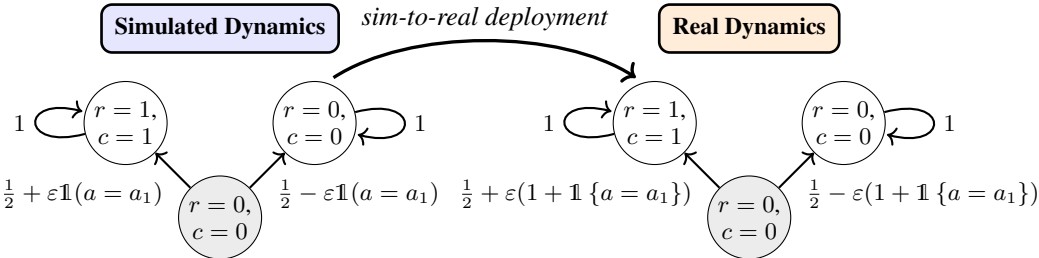

Figure 10: Pathological unsafe transfer. Light grey states denote the initial state $s_0$. Constraints are prone to be violated, even under a small mismatch between the simulated and the real system dynamics.

# B  Proofs

## B.1  Preliminaries

We first present a well-established lemma, the proof of which can be found in many works, e.g. Agarwal et al. (2021, Lemma 1.16).

**Lemma B.1** (Telescoping lemma). *Given a policy $\pi$, for different dynamics $p$ and $q$, let $g_{q,\pi,f}(s,a) \triangleq \mathbb{E}_{s'\sim q(\cdot|s,a)} f^{p,\pi}(s') - \mathbb{E}_{s'\sim p(\cdot|s,a)} f^{p,\pi}(s')$, where $f$ is used to overload the notation for the constraint function $V_c$. Then, we have*

$$C_q(\pi) - C_p(\pi) = \frac{\gamma}{1-\gamma}\mathbb{E}_{(s,a)\sim d_{q,\pi}} g_{q,\pi,C}(s,a).$$

Starting from this point, we assume the following Lipschitz continuity condition holds throughout the remainder of this section:

**Assumption B.2** (Continuity). For any $\pi \in \Pi^s$, the state cost value $V_c^{p,\pi}(s)$ and state reward value $V_r^{p,\pi}(s)$ are $L_C$- and $L_J$-Lipschitz continuous in $s \in \mathcal{S}$ w.r.t. the 1-norm respectively, over both $\mathcal{M}^\star$ and $\widehat{\mathcal{M}}_\xi$ for all $\xi \in \Xi$.

**Assumption B.3** (Finite Discrepancy). We assume that the worst-case model discrepancy between the real and all simulated environments for all $(s,a) \in \mathcal{S} \times \mathcal{A}$ is finite, i.e.,

$$\max_{\xi\in\Xi} D_W(\hat{p}_\xi, p^\star)(s,a) < \infty, \ \forall (s,a) \in \mathcal{S} \times \mathcal{A}.$$

## B.2  Proof of Lemma 4.1

*Proof.* Given a policy $\pi$, for dynamics $p^\star$ and $\hat{p}_\xi$, let $g_{\hat{p}_\xi,\pi,C}(s,a) \triangleq \mathbb{E}_{s'\sim\hat{p}_\xi(\cdot|s,a)} V_c^{p^\star,\pi}(s') - \mathbb{E}_{s'\sim p^\star(\cdot|s,a)} V_c^{p^\star,\pi}(s')$. Let $||\cdot||_{Lip}$ denote the Lipschitz constant of a function. Under the assumption that the state cost function is Lipschitz, by the Kantorovich-Rubinstein representation (Dudley, 2002, Section 11.8) for $L_1$ Wasserstein distance, we have:

$$g_{\hat{p}_\xi,\pi,C} \leq L_C \sup_{||f||_{Lip}\leq 1} |\mathbb{E}_{s'\sim p^\star(\cdot|s,a)} f(s') - \mathbb{E}_{s'\sim\hat{p}_\xi(\cdot|s,a)} f(s')|$$
$$\leq L_C \cdot D_W(\hat{p}_\xi, p^\star)(s,a). \tag{7}$$

By Lemma B.1, we have

$$\begin{aligned}
\left|C_{p^\star}(\pi) - \mathbb{E}_{\xi\sim\mu} C_{\hat{p}_\xi}(\pi)\right| &\leq \left|\mathbb{E}_{\xi\sim\mu}[C_{p^\star}(\pi) - C_{\hat{p}_\xi}(\pi)]\right| \\
&= \left|\frac{\gamma}{1-\gamma}\mathbb{E}_{\xi\sim\mu}\left[\mathbb{E}_{(s,a)\sim d_{\hat{p}_\xi,\pi}}[g_{\hat{p}_\xi,\pi,C}(s,a)]\right]\right| && \text{(Lemma B.1)} \\
&\leq \frac{\gamma L_C}{(1-\gamma)}\mathbb{E}_{\xi\sim\mu}\left[\mathbb{E}_{(s,a)\sim d_{\hat{p}_\xi,\pi}}[D_W(\hat{p}_\xi, p^\star)(s,a)]\right]. && \text{(Equation (7))}
\end{aligned}$$

This ends the proof of Lemma 4.1. $\qquad\square$

## B.3 Proof of Theorem 4.2

*Proof.* We first prove that $\mathbb{E}_{\xi\sim\mu}[\tilde{C}_{\hat{p}_\xi}(\pi)] \geq C_{p^\star}(\pi)$ for any policy $\pi \in \Pi^s$. By definition of the penalized cost $\tilde{C}_{\hat{p}_\xi}(\pi)$ and using Lemma 4.1, we have

$$
\begin{aligned}
&\mathbb{E}_{\xi\sim\mu}[\tilde{C}_{\hat{p}_\xi}(\pi)] - C_{p^\star}(\pi) \\
&= \mathbb{E}_{\xi\sim\mu}[C_{\hat{p}_\xi}(\pi)] - C_{p^\star}(\pi) \\
&\quad + \mathbb{E}_{\xi\sim\mu}\left[\frac{\gamma L_C}{1-\gamma}\mathbb{E}_{(s,a)\sim d_{\hat{p}_\xi},\pi}[\max_{\xi'\in\Xi} D_W(\hat{p}_{\xi'}, p^\star)(s,a)]\right] \\
&\geq -\frac{\gamma L_C}{(1-\gamma)}\mathbb{E}_{\xi\sim\mu}\left[\mathbb{E}_{(s,a)\sim d_{\hat{p}_\xi},\pi}[D_W(\hat{p}_\xi, p^\star)(s,a)]\right] \\
&\quad + \frac{\gamma L_C}{(1-\gamma)}\mathbb{E}_{\xi\sim\mu}\left[\mathbb{E}_{(s,a)\sim d_{\hat{p}_\xi},\pi}[\max_{\xi'\in\Xi} D_W(\hat{p}_{\xi'}, p^\star)(s,a)]\right] \quad \text{(Lemma 4.1)} \\
&\geq 0. \tag{8}
\end{aligned}
$$

Where in the last inequality we used that $D_W(\hat{p}_\xi, p^\star)(s,a) \leq \max_{\xi'\in\Xi} D_W(\hat{p}_{\xi'}, p^\star)(s,a)$. Next, since $\mathbb{E}_{\xi\sim\mu}[\tilde{C}_{\hat{p}_\xi}(\tilde{\pi})] \leq d$, and with Equation (Lemma 4.1) directly gives us:

$$
C_{p^\star}(\tilde{\pi}) \leq \mathbb{E}_{\xi\sim\mu}[\tilde{C}_{\hat{p}_\xi}(\tilde{\pi})] \leq d.
$$

This ends the proof of Theorem 4.2. $\qquad\square$

## C  Designing $\upsilon(s, a)$

In this section we provide additional theoretical intuition for the variance estimator introduced in Equation (6). Next, we show that, under that under the assumptions stated below, $\upsilon(\cdot, \cdot)$ upper-bounds the true sim-to-real discrepancy, measured by the $L_1$-Wasserstein distance, with high probability, using data collected in simulation. Importantly, when $\upsilon(\cdot, \cdot)$ is indeed an upper bound for the true model discrepancy, SPiDR is *provably* guaranteed to be safe, as described above in Appendix B.

### C.1  Modeling Assumptions

Let $d(\cdot, \cdot)$ be the Euclidean 2-norm on $\mathcal{S}$, and let $p_\mu \triangleq \mathbb{E}_{\xi\sim\mu}[p_\xi]$ denote the *domain-randomization kernel*.

**Definition C.1** ($L_1$-Wasserstein distance (Givens and Shortt, 1984))**.** Given a $\sigma$-algebra $\mathcal{F}$, for any two probability measures $P_1, P_2 \in \mathcal{M}(\mathcal{S}, \mathcal{F})$, the $L_1$-Wasserstein distance between them is defined as:

$$
D_W(P_1, P_2) \triangleq \inf_{\gamma\in\Gamma(P_1,P_2)} \int_{\mathcal{S}\times\mathcal{S}} d(s_1, s_2)\gamma(ds_1 \times ds_2), \tag{9}
$$

where $\Gamma(P_1, P_2) \triangleq \{\gamma \in \mathcal{M}(\mathcal{S}\times\mathcal{S}, \mathcal{F}\times\mathcal{F}) : \gamma(A\times\mathcal{S}) = P_1(A), \gamma(\mathcal{S}\times A) = P_2(A), \forall A \in \mathcal{F}\}$ is the set of all couplings of $P_1$ and $P_2$, and $d(\cdot, \cdot)$ is a $\mathcal{F}\times\mathcal{F}$-measurable metric defined on $\Xi$. In our setting, we assume this metric is given by a 2-norm $||\cdot||_2$.

**Assumption C.2** (Mild sim-to-real gap)**.** There exists a constant $\varepsilon > 0$, such that for any $(s, a) \in \mathcal{S}\times\mathcal{A}$, $\text{KL}(p^\star, p_\mu) \leq \varepsilon$ and $\max_{\xi\in\Xi}\text{KL}(p_\xi, p_\mu) \leq \varepsilon$.

This assumption states that the real dynamics are close to the domain-randomization kernel and that the domain-randomization support $\Xi$ is not overly large. In practice, practitioners design $\Xi$ using the domain knowledge of the real system; the uniform KL radius formalizes this intuition. Additionally, assuming that $\text{KL}(p^\star, p_\mu) \leq \varepsilon$ ensures that the simulator provides a reasonably accurate approximation of the real system. While the Wasserstein distance $D_W$ serves as the primary metric for model mismatch in our analysis, we use the KL divergence as it yields tighter concentration bounds, which we use below.

**Assumption C.3** (Bounded state space)**.** The state space $\mathcal{S}$ is compact with diameter $d_s$.

This is a fairly mild assumption in many physical problems, common for example in robotics. We next assume the sample average distance to the average next state in Algorithm 1 is uniformly lower bounded.

**Assumption C.4** (Non-degenerate variance). Let $s_i$ be $n$ i.i.d. draws from $p_\mu(\cdot \mid s, a)$ and $\overline{s}_{s,a} \triangleq \frac{1}{n} \sum_{i=1}^n s_i$ their empirical average. Furthermore, let $\hat{p}_\mu(s' \mid s, a) \triangleq \frac{1}{n} \sum_{i=1}^n \mathbb{1}[s' = s_i]$ denote the empirical measure associated with the sample set. We assume the expected distance between $s_i$ and $\overline{s}_{s,a}$ is uniformly lower bounded by a constant $c_3$, i.e.,

$$\mathbb{E}_{s' \sim \hat{p}_\mu(\cdot \mid s, a)}[d(s', \overline{s}_{s,a})] \geq c_3, \quad \forall (s, a) \in \mathcal{S} \times \mathcal{A}.$$

If the next-state distribution collapses to a point ($c_3 \approx 0$) the empirical variance trivially underestimates any model mismatch. Requiring a minimal spread rules out this degenerate case. In addition, many real systems are not fully deterministic and exhibit process and sensor noise. Hence a uniform lower bound $c_3$ is both mild and practically satisfiable.

### C.2 Auxiliary Lemmas

The following lemmas are used in our analysis below.

**Lemma C.5** (Bernstein transportation (Talebi and Maillard, 2018)). *Let $p, q \in \Sigma_S$, where $\Sigma_S$ denotes the probability simplex of dimension $S - 1$. For all $\alpha > 0$, for all functions $f$ defined on $\mathcal{S}$ with $0 \leq f(k) \leq b$, for all $s \in \mathcal{S}$, if $\mathrm{KL}(p, q) \leq \alpha$ then*

$$|pf - qf| \leq \sqrt{2\mathrm{Var}_q(f)\alpha} + \frac{2}{3}b\alpha,$$

*where we use the expectation operator defined as $pf \triangleq \mathbb{E}_{s \sim p} f(s)$ and the variance operator defined as $\mathrm{Var}_p(f) \triangleq \mathbb{E}_{s \sim p}(f(s) - \mathbb{E}_{s' \sim p} f(s'))^2 = p(f - pf)^2$.*

**Lemma C.6** (Variance bound for change of measure (Ménard et al., 2021)). *Let $p, q \in \Sigma_S$ and $f$ is a function defined on $S$ such that $0 \leq f(s) \leq b$ for all $s \in S$. If $\mathrm{KL}(p, q) \leq \beta$ then*

$$\mathrm{Var}_q(f) \leq 2\mathrm{Var}_p(f) + 4b^2\beta \quad \text{and} \quad \mathrm{Var}_p(f) \leq 2\mathrm{Var}_q(f) + 4b^2\beta.$$

**Lemma C.7** (Jonsson et al. (2020, Proposition 1)). *For all $p \in \Sigma_m$ and for all $\alpha \in [0, 1]$,*

$$\mathbb{P}\left(\forall n \in \mathbb{N}^+, n\mathrm{KL}(\hat{p}_n, p) \leq \log\left(\frac{1}{\alpha}\right) + (m-1)\log\left(e\left(1 + \frac{n}{m-1}\right)\right)\right) \geq 1 - \alpha.$$

### C.3 A High-probability Wasserstein bound

We are now ready to state the theoretical guarantee for $\upsilon(\cdot, \cdot)$, showing that it upper-bounds the worst-case $L_1$-Wasserstein distance between the simulated and real dynamics.

**Theorem C.8.** *Under Assumptions C.2-C.4, given a confidence level $\alpha \in (0, 1)$, for any $(s, a) \in \mathcal{S} \times \mathcal{A}$, there exist constants $C_1, C_2, C_3, C_4 > 0$ that contain only $S, A, \alpha, d_s, \varepsilon$ and $\log$ factor of $n$, such that with probability at least $1 - \alpha$, we have:*

$$\max_{\xi \in \Xi} D_W(p_\xi, p^\star)(s, a) \leq C_1 \upsilon(s, a) + C_2\varepsilon + \sqrt{\frac{C_3}{n}} + \frac{C_4}{n}.$$

The leading term $C_1\upsilon(s, a)$ captures how the *local* spread of simulated transitions bounds the worst-case Wasserstein gap. The $\varepsilon$ term reflects irreducible model mismatch, and the remaining terms are standard finite-sample corrections. The key insight lies in the first term, which uses the variance in our algorithm. We note that while the design of $\upsilon(\cdot, \cdot)$ is not necessarily the tightest possible, it is easy to implement, making it more widely applicable. Our proof for Theorem C.8 is stated below.

*Proof.* We first establish an upper bound on $D_W(p_\xi, p^\star)(s, a)$ using the 2-norm. Specifically, for each $(s, a)$, consider the coupling between $p_\xi(\cdot \mid s, a)$ and $p^\star(\cdot \mid s, a)$, defined by:

$$\gamma(s_1, s_2)_{s,a} = p_\xi(s_1 \mid s, a) \cdot p^\star(s_2 \mid s, a).$$

By the definition of the $L_1$-Wasserstein distance given in Equation (9), we have:

$$D_W(p_\xi, p^\star)(s, a) \leq \mathbb{E}_{s_1 \sim p_\xi(\cdot \mid s, a)}\left[\mathbb{E}_{s_2 \sim p^\star(\cdot \mid s, a)}[d(s_1, s_2)]\right]. \tag{10}$$

Recall that $\bar{s}_{s,a} \triangleq \frac{1}{n} \sum_{i=1}^{n} s_i$, where $s_i$ are i.i.d. drawn from $p_\mu(\cdot \mid s, a)$. By the triangle inequality of the 2-norm, we have:

$$d(s_1, s_2) \leq d(s_1, \bar{s}_{s,a}) + d(s_2, \bar{s}_{s,a}).$$

Substituting this into Equation (10) yields:

$$D_W(p_\xi, p^\star)(s, a) \leq \mathbb{E}_{s_1 \sim p_\xi(\cdot \mid s, a)}[d(s_1, \bar{s}_{s,a})] + \mathbb{E}_{s_2 \sim p^\star(\cdot \mid s, a)}[d(s_2, \bar{s}_{s,a})]. \tag{11}$$

Since $p_\xi$ and $p^\star$ are unknown, we approximate them using $p_\mu$ via Lemma C.5. Under Assumption C.2 and Assumption C.3, we obtain:

$$\mathbb{E}_{s_1 \sim p_\xi(\cdot \mid s, a)}[d(s_1, \bar{s}_{s,a})] \leq \mathbb{E}_{s_1 \sim p_\mu(\cdot \mid s, a)}[d(s_1, \bar{s}_{s,a})] + \sqrt{2\mathrm{Var}_{p_\mu}(d(s_1, \bar{s}_{s,a}))\varepsilon} + \frac{2}{3} d_s \cdot \varepsilon,$$

$$\mathbb{E}_{s_2 \sim p^\star(\cdot \mid s, a)}[d(s_2, \bar{s}_{s,a})] \leq \mathbb{E}_{s_2 \sim p_\mu(\cdot \mid s, a)}[d(s_2, \bar{s}_{s,a})] + \sqrt{2\mathrm{Var}_{p_\mu}(d(s_2, \bar{s}_{s,a}))\varepsilon} + \frac{2}{3} d_s \cdot \varepsilon.$$

Substituting into Equation (11) gives:

$$D_W(p_\xi, p^\star)(s, a) \leq 2\mathbb{E}_{s' \sim p_\mu(\cdot \mid s, a)}[d(s', \bar{s}_{s,a})] + 2\sqrt{2\mathrm{Var}_{p_\mu}(d(s', \bar{s}_{s,a}))\varepsilon} + \frac{4}{3} d_s \cdot \varepsilon \tag{12}$$

Recall that $\hat{p}_\mu(s' \mid s, a) \triangleq \frac{1}{n} \sum_{i=1}^{n} \mathbb{1}[s' = s_i]$ denote the empirical measure. By Lemma C.7, we have w.p. $1 - \alpha$, for any $(s, a)$, the following inequality holds:

$$\mathrm{KL}(\hat{p}_\mu, p_\mu)(s, a) \leq \frac{1}{n} \left( \log\left(\frac{1}{\alpha}\right) + (S-1)\log\left(e\left(1 + \frac{n}{S-1}\right)\right) \right)$$

Define $g(S, A, n, \alpha) \triangleq \log\left(\frac{1}{\alpha}\right) + (S-1)\log\left(e\left(1 + \frac{n}{S-1}\right)\right)$. Applying Lemma C.5 and Lemma C.6 to the terms appearing in Equation (12) gives:

$$\mathbb{E}_{s' \sim p_\mu(\cdot \mid s, a)}[d(s', \bar{s}_{s,a})] \leq \mathbb{E}_{s' \sim \hat{p}_\mu(\cdot \mid s, a)}[d(s', \bar{s}_{s,a})] + \sqrt{2\mathrm{Var}_{\hat{p}_\mu}[d(s', \bar{s}_{s,a})]\frac{g(S, A, n, \alpha)}{n}}$$

$$+ \frac{2d_s g(S, A, n, \alpha)}{3n} \qquad \text{(Lemma C.5)}$$

$$\leq \mathbb{E}_{s' \sim \hat{p}_\mu(\cdot \mid s, a)}[d(s', \bar{s}_{s,a})] + \sqrt{\frac{2d_s^2 g(S, A, n, \alpha)}{n}} + \frac{2d_s g(S, A, n, \alpha)}{3n},$$
$$(\mathrm{Var}_p[f] \leq \mathbb{E}_p[f^2])$$

and

$$\sqrt{2\mathrm{Var}_{p_\mu}(d(s', \bar{s}_{s,a}))\varepsilon} \leq 2\sqrt{\mathrm{Var}_{\hat{p}_\mu}(d(s', \bar{s}_{s,a}))\varepsilon + 2d_s^2 \varepsilon \frac{g(S, A, n, \alpha)}{n}} \qquad \text{(Lemma C.6)}$$

$$\leq 2\sqrt{\mathrm{Var}_{\hat{p}_\mu}(d(s', \bar{s}_{s,a}))\varepsilon} + 2\sqrt{2d_s^2 \varepsilon \frac{g(S, A, n, \alpha)}{n}}.$$
$$(\sqrt{x+y} \leq \sqrt{x} + \sqrt{y})$$

Substituting these into Equation (12) yields that with probability at least $1 - \alpha$, for any $(s, a)$ and any $\xi \in \Xi$, the following inequality holds:

$$
\begin{aligned}
D_W(p_\xi, p^\star)(s, a) &\leq 2\mathbb{E}_{s' \sim \hat{p}_\mu(\cdot|s,a)}[d(s', \bar{s}_{s,a})] + 2\sqrt{\frac{2d_s^2 g(S, A, n, \alpha)}{n}} + \frac{4d_s g(S, A, n, \alpha)}{3n} \\
&\quad + 4\sqrt{\mathrm{Var}_{\hat{p}_\mu}(d(s', \bar{s}_{s,a}))\varepsilon} + 4\sqrt{2d_s^2 \varepsilon \frac{g(S, A, n, \alpha)}{n}} + \frac{4}{3}d_s \cdot \varepsilon \\
&\leq \frac{2}{c_3}(\mathbb{E}_{s' \sim \hat{p}_\mu(\cdot|s,a)}[d(s', \bar{s}_{s,a})])^2 + 2\sqrt{\frac{2d_s^2 g(S, A, n, \alpha)}{n}} + \frac{4d_s g(S, A, n, \alpha)}{3n} \\
&\quad + 4\sqrt{\mathrm{Var}_{\hat{p}_\mu}(d(s', \bar{s}_{s,a}))\varepsilon} + 4\sqrt{2d_s^2 \varepsilon \frac{g(S, A, n, \alpha)}{n}} + \frac{4}{3}d_s \cdot \varepsilon \\
&\hspace{6cm} \text{(Assumption C.4)} \\
&\leq \frac{2}{c_3}(\mathbb{E}_{s' \sim \hat{p}_\mu(\cdot|s,a)}[d(s', \bar{s}_{s,a})])^2 + 2\sqrt{\frac{2d_s^2 g(S, A, n, \alpha)}{n}} + \frac{4d_s g(S, A, n, \alpha)}{3n} \\
&\quad + \frac{2}{c_3}\mathrm{Var}_{\hat{p}_\mu}(d(s', \bar{s}_{s,a})) + 4\sqrt{2d_s^2 \varepsilon \frac{g(S, A, n, \alpha)}{n}} + (\frac{4}{3}d_s + 2c_3) \cdot \varepsilon \\
&\hspace{6cm} \text{(Young's inequality)} \\
&= \frac{2}{c_3}\upsilon(s, a) + 2\sqrt{\frac{2d_s^2 g(S, A, n, \alpha)}{n}} + \frac{4d_s g(S, A, n, \alpha)}{3n} \\
&\quad + 4\sqrt{2d_s^2 \varepsilon \frac{g(S, A, n, \alpha)}{n}} + (\frac{4}{3}d_s + 2c_3) \cdot \varepsilon, \hspace{2cm} (13)
\end{aligned}
$$

where the last equality utilizes that $\mathbb{E}[f^2] = (\mathbb{E}[f])^2 + \mathrm{Var}[f]$ and the fact that

$$
\upsilon(s, a) = \sum_{j=1}^{\dim(\mathcal{S})} \mathrm{Var}(s_{1,j}, \ldots, s_{n,j}) = \mathbb{E}_{s' \sim \hat{p}_\mu(\cdot|s,a)}\left[[d(s', \bar{s}_{s,a})]^2\right]
$$

This ends the proof. $\hfill\square$

# D   Additional Sim-to-Sim Ablations

We conduct further empirical analysis of SPiDR, focusing on the following key aspects: **(i)** sensitivity to the penalty parameter $\lambda$, **(ii)** further analysis of standard domain randomization, **(iii)** empirical validation of Lemma 4.1 and, **(iv)** behavior of the uncertainty approximation $v(s, a)$ Through these ablations, we highlight SPiDR's robustness to the choice of $\lambda$, its efficiency in scaling to larger domain ensembles, and its principled handling of uncertainty in challenging state-action regions.

**How robust is SPiDR to the choice of $\lambda$?**   We study the performance of SPiDR across varying values of $\lambda$ and under different magnitudes of distribution shifts in the CartpoleBalance task. To this end, we vary the magnitude of the actuator's gear parameter, denoted as $|\Xi|$, with a slight abuse of notation. We ablate $\lambda \in \{0.85, 0.95, 0.75, 0.6, 0.5, 0.25, 0.35, 0.1, 0\}$ and $|\Xi|$ across $\{350, 400, 500, 500\}$. We present our results in Figure 11, where we compare the objective and constraint for different $\lambda$ values. Notably, for $\lambda = 0.6$, while some performance is sacrificed in the objective, depending on the magnitude of $|\Xi|$, the constraint is satisfied across all settings of $|\Xi|$.

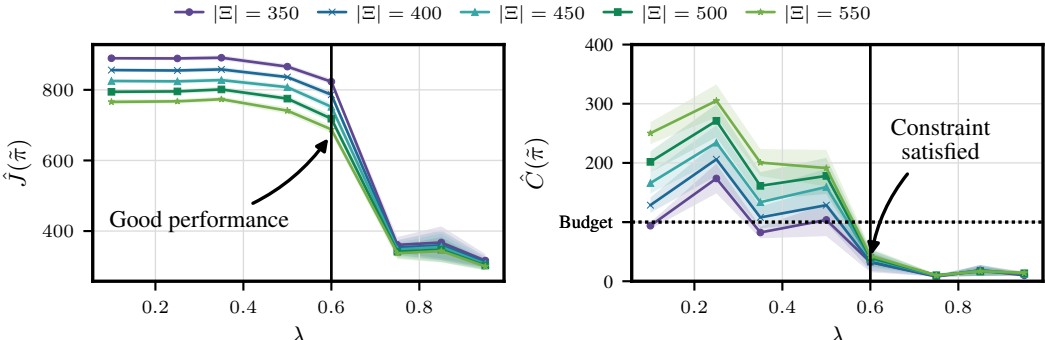

Figure 11: SPiDR's performance under different values of motor gear parameter. One choice of $\lambda$ consistently maintains constraint satisfaction.

**Can domain randomization still be safe if we enlarge the set $\Xi$?**   We study the dependence of standard domain randomization on the size of the set $\Xi$, presenting an example where even significantly increasing this set, and therefore the "diversity" of training environments, fails to enable safe transfer. To this end, we use the simulated RaceCar environment and vary the lower and upper limits of the car's throttle parameter. This parameter controls how fast can the car drive, directly relating to its ability to operate safely upon deployment. Specifically, we use the following values $\{(0.4, 0.6), (0.3, 0.7), (0.2, 0.8), (0.1, 0.9)\}$, where each tuple defines the minimum and maximum throttle values. The default range used in our experiments (including sim-to-real experiments) is $(0.4, 0.6)$. As in the previous experiment, we slightly abuse notation and refer to the diameter of $\Xi$ as $|\Xi|$, computed as the difference between the upper and lower bounds. We report our results in Figure 12, demonstrating that for this task, safety upon transfer is not maintained, even when using

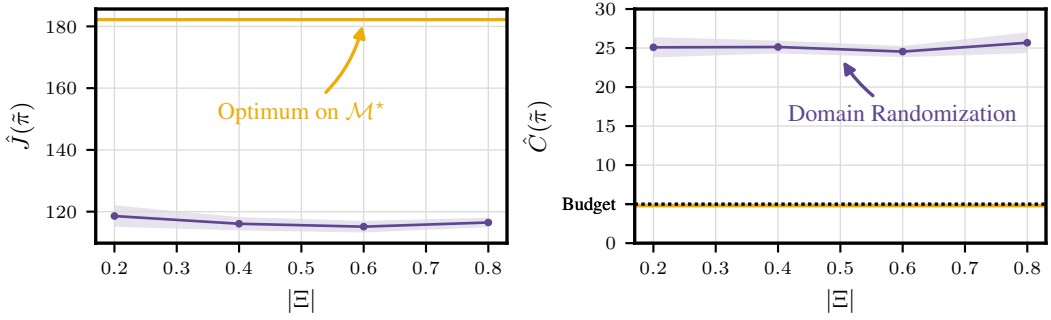

Figure 12: Constraint and objective of domain randomization when varying the size of the car's throttle parameter. Constraints are violated even when increasing the range of paramteres.

a wide range of parameters with domain randomization. More generally, we argue that for some

problems, simply enlarging $\Xi$ could generally improve safety, however when safety must be ensured, pessimism is necessary.

**Can we quantify constraint underestimation?** We investigate the degree of constraint underestimation from Lemma 4.1. To this end, we measure the difference in the constraint measured in the training environment against the evaluation environment. Concretely, we collect empirical estimates of $C_\star(\tilde{\pi})$ and $\mathbb{E}_{\xi \sim \mu} C_{\hat{p}_\xi}(\tilde{\pi})$ and report $C_\star(\pi) - \mathbb{E}_{\xi \sim \mu} C_{\hat{p}_\xi}(\tilde{\pi})$ for SPiDR and standard domain randomization in Figure 13. As shown, in all tasks, compared to SPiDR, standard domain randomization underestimates the cost, leading to unsafe behavior on the test tasks (cf. Figure 6).

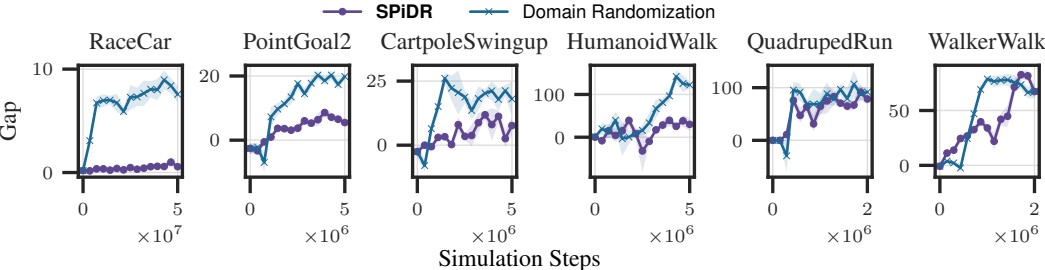

Figure 13: Constraint performance gap between training and test (lower is better). Domain randomization underestimates the constraint on the test tasks.

**Does $v(s, a)$ upper-bounds the discrepancy in practice?** We continue our previous study and demonstrate that $\mathbb{E}_{(s,a) \sim d_{\hat{p}_\xi}, \pi}[\lambda v(s, a)] - |C_\star(\pi) - \mathbb{E}_{\xi \sim \mu} C_{\hat{p}_\xi}(\pi)| \gtrsim 0$, namely, that our estimate $v(\cdot, \cdot)$ can be used to sufficiently penalize the cost. In Figure 14 we report this error across all six simulated tasks. As shown, in all tasks, our approximation error is positive, suggesting that $\lambda v$ indeed

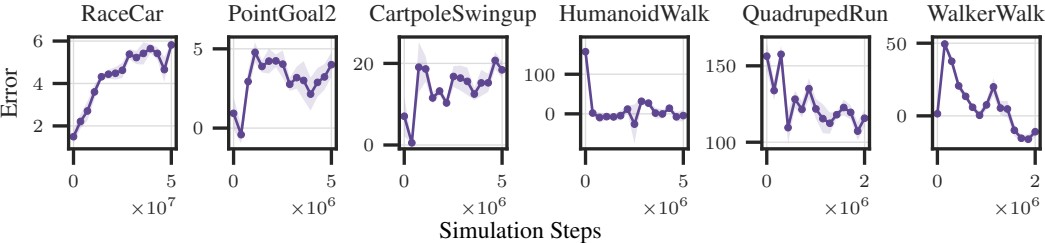

Figure 14: Our estimator $v(s, a)$ upper bounds the model discrepancy.

upper-bounds the gap in constraint. Notably, in the HumanoidWalk environment, the error of $\lambda v(s, a)$ w.r.t. the constraint gap is close to zero. This result is in line with the performance-safety tradeoff shown in Figure 7, were SPiDR is both safe and achieving strong performance compared to the best solution on $\mathcal{M}^\star$.

**How does $v(s, a)$ vary across states?** We analyze $v(s, a)$ measured across different states and actions in the CartpoleSwingup environment. The state space consists of the linear cart position and velocity, along with the angular position ($\theta$) and angular velocity ($\dot{\theta}$) of the pole. The continuous action space lies in $[-1, 1]$, and due to the symmetry of the system, we restrict our analysis to $a \in [0, 1]$. Figure 15 visualizes the uncertainty over the angular position and velocity dimensions for representative actions $a \in \{0.0, 0.3, 0.7, 1.0\}$. We note the following observations: **(i)** uncertainty generally increases with action magnitude, and, **(ii)** for $a \neq 0$, uncertainty peaks when the pole is near the upright position ($\theta \approx \pi$), which corresponds to an inherently unstable equilibrium. This observation aligns with intuition: near the unstable upright position, small perturbations can lead to significantly different outcomes, making these regions harder to model. From a safety perspective, this analysis motivates penalizing high-uncertainty regions during policy learning, particularly when deploying on real systems where model inaccuracies may have significant consequences.

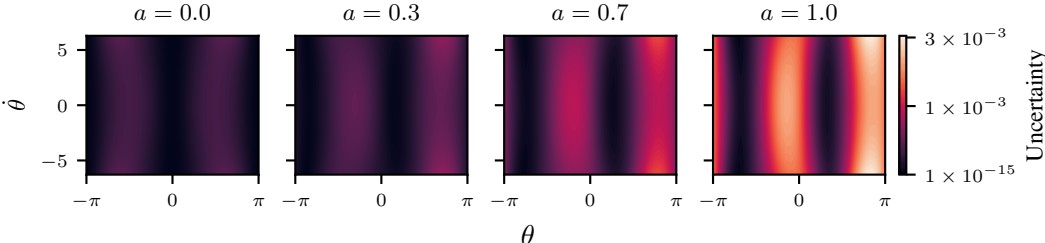

Figure 15: Uncertainty $\upsilon(s, a)$ across angular states for actions $a \in \{0.0, 0.3, 0.7, 1.0\}$. Uncertainty increases with action magnitude and is highest near the upright position ($\theta \approx \pi$), highlighting states where the simulator's predictions are less reliable.

# E  Picking $\lambda$ in Practice

We demonstrate our ablations on $\lambda$ when deploying SPiDR in real, providing guidance on how to pick it effectively in practice. We describe below the procedure we use in our sim-to-real experiments for selecting $\lambda$.

Step 1: **Estimate the magnitude of $\upsilon(\cdot, \cdot)$.** Evaluate and record $\upsilon(\cdot, \cdot)$ in training, aggregated over state-action pairs and across time. This estimate is not required to be precise, its purpose is to determine the general order of magnitude. This can be done purely in simulation.

Step 2: **Select an appropriate range for $\lambda$.** Choose a candidate range of $\lambda$ values such that $1/\upsilon \approx 1$. This heuristic ensures that the penalty term has roughly the same magnitude of cost function, assuming $c_{\max} \approx 1$.

Step 3: **Iteratively refine $\lambda$.** On the real system, begin testing with the largest value in the selected range and iteratively decrease it until the constraint satisfaction on the real system closely matches the desired budget.

In Figure 16 we report the performance of SPiDR across different values of $\lambda$. Importantly, for a

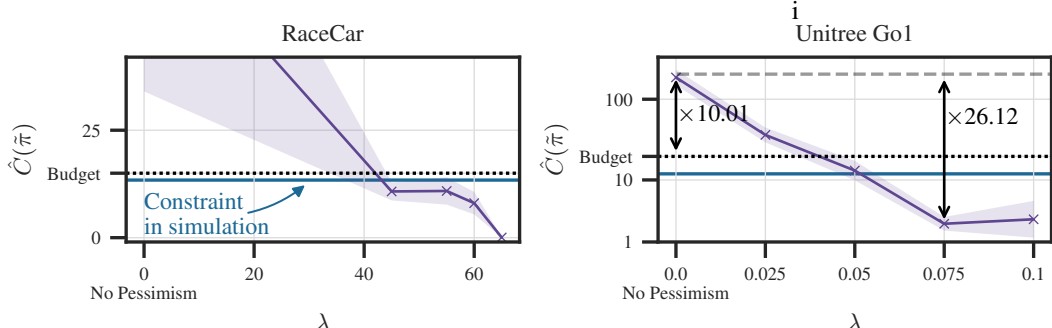

Figure 16: Safety performance under different $\lambda$ values in real-world robotic tasks. The blue horizontal line represents the value of the constraint measured *in simulation* under $\lambda = 0$. SPiDR satisfies the constraints on both of the physical systems.

large enough initial choice of $\lambda$, which can be obtained in practice using prior domain knowledge (e.g. degree of of simulator fidelity), the very first deployment of SPiDR is safe. In Step 3, online data is used to improve *performance*, while maintaining safety. With this procedure, safety is maintained zero-shot, while performance can be improved with online rollouts, in line with our safety guarantee in Section 4.3.

# F  Learning Curves for Simulated Environments

In Figure 17 we provide the full learning curves of the experiment trials used for Section 5.2, including the standard error intervals across five random seeds.

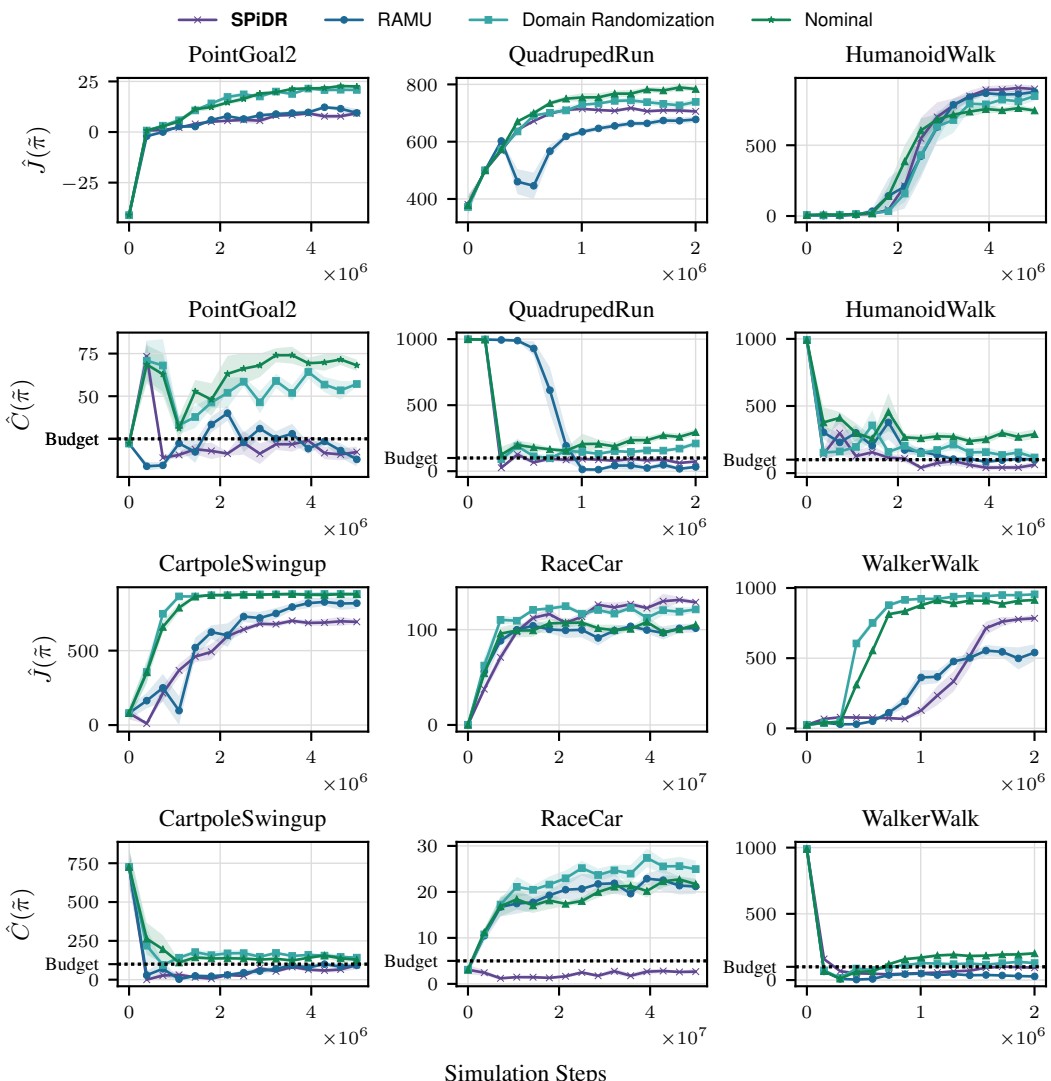

Figure 17: Learning curves used in Section 5.2. SPiDR consistently satisfies the constraints while maintaining good performance on the objective. Domain Randomization and Nominal fail to satisfy the constraints.

**Simulating the sim-to-sim gap.** We simulate the sim-to-sim gap as follows. In PointGoal2 and the RWRL tasks, we follow a similar approach to Queeney and Benosman (2024) and introduce in evaluation an additional dynamics parameter (e.g., mass or motor gains) that is not encountered during training. In the RaceCar environment, the agent is trained using a simplified bicycle model, but evaluated on a more realistic variant that incorporates tire forces and detailed motor dynamics. Further details on the tasks and their sim-to-sim gap design can be found in Appendices J to L.

## G   Locomotion Experiments with Unitree Go1

We train policies using the FlatTerrainGo1Joystick environment from MuJoCo Playground (Zakka et al., 2025). In this environment, the agent is tasked with following randomly sampled velocity commands in the forward, lateral, and yaw directions. Excessive joint motion can cause the legs to self-collide, even when joint limits are not reached. These collisions can lead to falls or serious hardware damage.

**Constraints.**   To prevent these joint limit violations we define a cost function to measure the number of joint violations. The cost function is defined by the indicator function of any joint being outside the soft limit. More formally, the cost is defined as

$$c(s, a) \triangleq \begin{cases} 1 & \text{if any of } j \in \mathcal{J} \text{ such that } q_j > 0.75 \cdot q_j^{\max} \text{ or } q_j < 0.75 \cdot q_j^{\min} \\ 0 & \text{otherwise} \end{cases}$$

where $\mathcal{J}$ is the set of joint indices, $q_j$ is the joint angle and $q_j^{\max}, q_j^{\min}$ are the angle limits of joint angle $j$. The soft factor limit of 0.75 is applied meaning that if a joint enters the outermost 25% of its feasible range of motion, the cost is set to 1. We choose this constraint since joint-position measurements are accurate and reliable on the real system, without relying on indirect filtering/estimation methods.

**Training in simulation.**   Each policy is trained for roughly one hour on an NVIDIA RTX4090 GPU. For both SAC and PPO, we train policies with a primal-dual solver, using different values of $\lambda$ to penalize the uncertainty. Each $\lambda$ value is trained across five different random seeds.

**Command distribution.**   Target commands are uniformly sampled from the ranges $[\pm 0.45, \pm 0.2, \pm 1.3]$, corresponding to the forward velocity, lateral velocity, and yaw rate, respectively. Each sampled command is applied for a fixed number of 1000 control steps. After this period, the command is reversed, perturbed with additive noise, and then reapplied for the same duration, after which a new command is sampled from the same distribution as stated above.

**Real-world evaluation.**   We evaluate the trained policies on the Unitree Go1 quadruped robot. To ensure that all policies are evaluated under identical conditions, the same sequences of commands are sampled for each episode. Every policy is tested across 10 independent trials. We provide .onnx files for the policies used in this experiment, as well as the code to port simulation-trained policies to .onnx format in the following link.

**Comparison with** RAMU.   In Figure 18 we demonstrate a comparison of SPiDR and RAMU, conditioned on the same sequence of commands. We note that RAMU uses only a single "nominal" environment, together with robust estimation value function estimation as means for robust transfer. We believe that RAMU fails to follow the commands mainly because using only the nominal environment, without incorporating domain randomization, might not be sufficient for robust sim-to-real transfer. We note that conceptually RAMU and domain randomization are not mutually exclusive, therefore RAMU could in principle transfer better if used with domain randomization. However, combining the two goes beyond the proposed algorithmic solution and official implementation of RAMU, and therefore not considered in this experiment. Despite that, we believe that combining the two approach is an interesting direction for future work.

## H   RaceCar Experiments

In this task we implement the simulated RaceCar environment on a real remote-controlled car, illustrated in Figure 2. Please see Appendix L for further details about the reward and cost functions.

**Real-world implementation.**   We measure the position and orientation of the car using a motion-capture system. Velocities are estimated using a first-order low pass filter. These measurements are sufficient to recover the full state of the system as the goal and obstacle positions are fixed. All trajectory measurements start from the roughly same initial position in the world frame. The goal position is at the origin. Further details, regarding the parameters used for domain randomization and training hyper-parameters, can be found in our open-source implementation.

**Additional results.**   In Figure 19 we provide the objective and constraint measured on the real system in this experiment. As shown, for $\lambda = 45$, SPiDR is able to satisfy the constraint, while finding a performant policy.

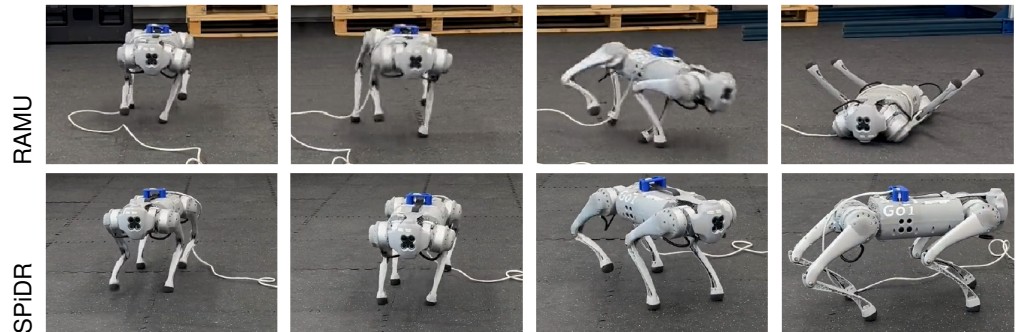

Figure 18: Comparison of RAMU and SPiDR on the Unitree Go1 robot. SPiDR follows the given commadns without falling.

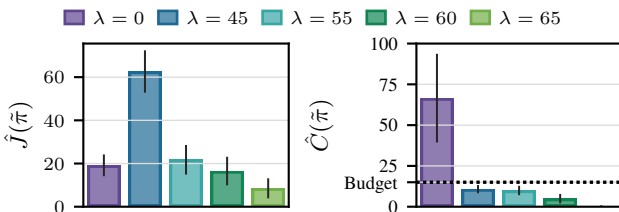

Figure 19: Safety and objective performance for $\lambda \in \{0, 45, 55, 60, 65\}$ on the real system. We report the mean and standard error across five seeds. SPiDR transfers safely to the real system while solving the task, i.e., reaching to the goal position.

**Comparison with constraint tightening.** We additionally compare SPiDR with a simple baseline that tightens the constraint budget $d$ in simulation. To this end, we evaluate "constraint tightening" on the real-world RaceCar, ablating it when using budgets $d \in \{0, 7.5\}$ when training in simulation. We present our results in Figure 20. As shown, while reducing the budget $d$ in simulation indeed reduces the degree of accumulated costs on the real system, it might still not be sufficient to maintain safety, opposed to SPiDR. While this approach is even simpler than SPiDR, there are two main

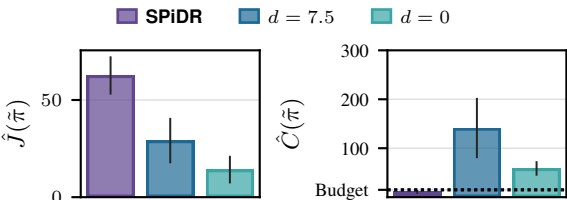

Figure 20: Comparison of SPiDR with constraint tightening on the real RaceCar. Constraint tightening fails to satisfy the constraint.

challenges with it. First, severe tightening may yield a "zero budget" training regime that CMDP solvers like CRPO and primal-dual methods struggle to solve in practice (Huang et al., 2023; He et al., 2023). This behavior is observerd in Figure 20; even though the budget is set to $d = 0$, the primal-dual CMDP solver we use fails to converge to a constraint satisfying solution. Second, as hinted, reducing the budget is equivalent to finding a uniform upper-bound to the penalty term in Equation (4), that does not depend on state-actions. Therefore, in order to achieve the same level of penalty required to satisfy the constraint, this for of "uniform pessimism" can degrade performance by being over-pessimistic in those states where the uncertainty about the sim-to-real gap might in fact be low, effectively penalizing the "wrong" states. We visualize such state-action-dependent uncertainty in Figure 15.

# I  Additional Experiments with PPO

To further study SPiDR's performance with CMDP solvers that utilize policy gradients, we use PPO (Schulman et al., 2017). PPO is a common choice used in many robotics tasks (for instance, see Lee et al., 2020). We use SauteRL as a CMDP solver (Sootla et al., 2022). SuateRL augments the state with a counter of the online accumulated cost, and penalizes the reward once this counter exceeds the budget. We ablate $\lambda \in \{0, 0.001, 0.01, 1\}$ and report the performance on the CartpoleBalance task from RWRL in Figure 21. As shown, the constraints are violated for $\lambda = 0$, corresponding to

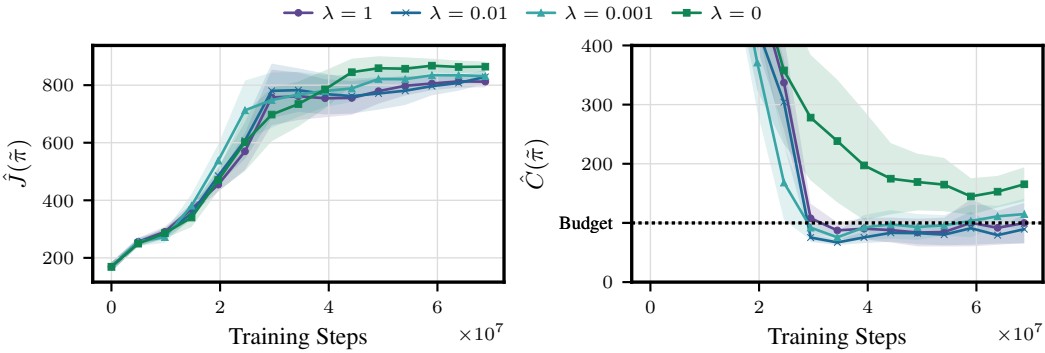

Figure 21: PPO with SauteRL as a CMDP solver. Domain randomization fails to satisfy the constraints, while for $\lambda \geq 0.01$ constraints are satisfied with minor performance drop.

standard domain randomization. In contrast, the constraints are satisfied as $\lambda$ increases. We note that, while SauteRL is designed for constraint satisfaction with high probability, instead of bounding the expectation of the cumulative costs, and thus in principle it adds additional conservatism, this might still not guarantee safe transfer under modeling mismatches. Figure 21 demonstrates that SPiDR mitigates this issue.

# J  Safety Gym

To compare policies in environments from OpenAI Safety Gym by Ray et al. (2019), we port the PointGoal2 environment from standard MuJoCo to MJX (MuJoCo XLA). MJX is a JAX-compiled MuJoCo backend that is tightly integrated with Brax (Freeman et al., 2021). This port enables us to massively parallelize training by collecting trajectories from thousands of environments in parallel on a single GPU, accelerating training by several orders of magnitude compared to the original implementation. This contribution is of independent interest to the safe RL community. Extending our work to additional Safety Gym environments is left for future work.

**PointGoal2 environment.**  In this environment, the agent must navigate to a target location while avoiding hazards which include free-moving vases and designated hazard zones. The environment is depicted in Figure 22. The initial positions of the agent, goal, vases, and hazards are randomized at the beginning of training. The environment reward is defined as the change in Euclidean distance to the goal between successive steps

$$r_t(s_t, a_t) \triangleq d_{t-1} - d_t + \mathbb{1}[d_t \leq \epsilon],$$

where $d_t = \|\mathbf{x}_t - \mathbf{x}_{\text{goal}}\|_2$ is the Euclidean distance from the robot to the goal. The term $\mathbb{1}[d_t \leq \epsilon]$ is an indicator function that gives a reward bonus when the agent reaches the goal, i.e., when it is within $\epsilon = 0.3$ of the center of the goal. The goal position is resampled to another free position in the environment once reached.

A cost of 1 is incurred when the agent collides with a vase $v$, when one of the vases crosses a linear velocity threshold (after collision), or when the agent is inside a hazard zone $h$:

$$c_t(s_t, a_t) \triangleq \mathbb{1}[\exists v \in V : \text{collides}(\mathbf{x}_t, \mathbf{x}_v)] + \mathbb{1}[\exists v \in V : \mathbf{x}'_v \geq \gamma] + \mathbb{1}[\exists h \in H : d_t \leq \rho],$$

where $\gamma = 5e^{-2}$ and $\rho = 0.2$. Please see the implementation of Ray et al. (2019) for more specific details and our open-source implementation.

**Sim-to-sim gap.** During evaluation, we uniformly sample three key parameters of the environment's dynamics: multiplicative joint damping and mass factors and additive values for the actuator gear ratio. The precise training and evaluation ranges are given in Table 1. Note that the $z$-joint is a hinge joint that allows the agent to rotate around the $z$-axis and the $x$-joint is a slide joint that allows translation in the $xy$-plane.

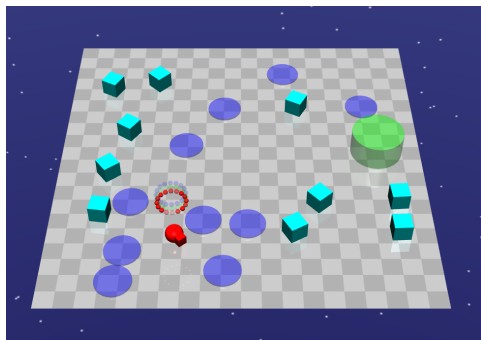

Figure 22: Visualization of a random initialization of the PointGoal2 environment. The red pointmass is the agent, the green transparent cylinder is the goal, the cyan boxes are vases and the blue circles are hazard zones.

Table 1: Domain randomization parameters and ranges used during training and evaluation. $+$ and $\times$ denote additive and multiplicative terms respectively. By fixing the damping and mass parameters in training, but not in evaluation, we simulate both lack of knowledge of $\xi^\star$, but also modeling mismatch due to imperfect simulation.

| Parameter | Train | Eval |
|---|---|---|
| Damping (x, y) $\times$ | 1.0 (fixed) | [0.6, 1.0] |
| Damping (z) $\times$ | 1.0 (fixed) | [0.7, 1.0] |
| Gear (x) $+$ | [–0.2, 0.2] | [0, 0.1] |
| Gear (z) $+$ | [–0.1, 0.1] | [0, 0.05] |
| Mass $\times$ | 1.0 (fixed) | [1.0, 1.05] |

## K   RWRL Benchmark

We evaluate SPiDR on four robotic tasks using the RWRL benchmark (Dulac-Arnold et al., 2020), which adds safety constraints and distribution shifts to DeepMind Control suite tasks. We build on MuJoCo Playground (Zakka et al., 2025), an MJX-based reimplementation that enables faster, parallelized training, by incorporating RWRL's modifications. See our open-source implementation for details.

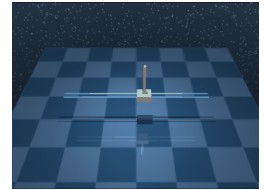 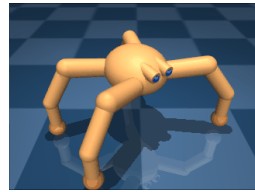 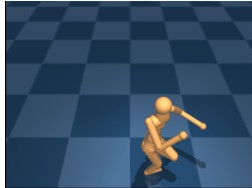 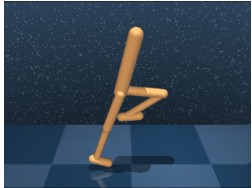

Figure 23: RWRL tasks.

**Constraints.** We use the joint position limits constraint for HumanoidWalk and QuadrupedRun, joint velocity limits for WalkerWalk, and slider position limits for CartpoleSwingup. These are the standard constraints proposed by Dulac-Arnold et al. (2020).

**Sim-to-sim gap.** We follow a similar experimental setup as Queeney and Benosman (2024) and introduce additional variability in the evaluation task to simulate modeling mismatches. In Table 2 we provide the specific parameters we perturb for the RWRL environments.

## L   RaceCar Environment

**Reward and cost.** The reward at timestep $t$ is given by

$$r_t(s_t, a_t) \triangleq d_{t-1} - d_t + \mathbb{1}[d_t \leq \epsilon] - \lambda_c \|a_t\|_2 - \lambda_l \|a_t - a_{t-1}\|_2^2,$$

where $d_t = \|\mathbf{x}_t - \mathbf{x}_{\text{goal}}\|_2$ is the Euclidean distance to the goal, and $a_t \in \mathbb{R}^2$ denotes the action applied at time $t$ (consisting of steering and throttle). The term $\mathbb{1}[d_t \leq \epsilon]$ is an indicator function that gives a reward bonus when the agent is within $\epsilon = 0.3$ of the goal. The penalties $\lambda_c$ and $\lambda_l$ weight the control effort (magnitude of the action) and the change in action between consecutive timesteps,

Table 2: Domain randomization parameters and ranges used during training and evaluation across tasks from RWLR. + and × denote additive and multiplicative terms respectively.

| Parameter | Train | Eval |
|---|---|---|
| **CartpoleSwingup** | | |
| Pole Length + | [0.0, 0.0] | [–0.25, 0.25] |
| Gear + | [0.0, 5.0] | [0.0, 5.0] |
| **QuadrupedRun** | | |
| Torso Density + | [0.0, 0.0] | [–300.0, 300.0] |
| Friction + | [0.0, 0.0] | [–0.95, 0.0] |
| Lift Gear × | [0.75, 1.5] | [0.75, 1.5] |
| Yaw Gear × | [1.0, 1.0] (fixed) | [0.5, 2.0] |
| Extend Gear × | [0.75, 1.5] | [0.75, 1.5] |
| **HumanoidWalk** | | |
| Friction + | [0.0, 0.0] | [-0.05, 0.05] |
| Hip Gear (x) + | [-20., 20.] | [-20., 20.] |
| Hip Gear (y) + | [-20., 20.] | [-20., 20.] |
| Hip Gear (z) + | [-60., 60.] | [-60., 60.] |
| Knee Gear + | [0, 0] | [-40., 40.] |
| **WalkerWalk** | | |
| Torso Length + | [0.0, 0.0] (fixed) | [–0.75, 0.75] |
| Gear + | [0.0, 20.0] | [0.0, 20.0] |

respectively. The cost function at time $t$ is defined as

$$c_t \triangleq \sum_{i=1}^{3} \mathbb{1}\left[\|x_t - p_i\| < \rho_i\right] E_t^k + \mathbb{1}[x_t \notin \mathcal{V}],$$

where $x_t \in \mathbb{R}^2$ is the agent's position, $p_i$ and $\rho_i$ are the position and radius of the $i$-th obstacle and $E_t^k$ is the kinetic energy of the car at time $t$, simulating a plastic collision between the car and obstacles. This choice of cost function allows us to penalize more severely collisions in which the car smashes into obstacles, as opposed to softly touching them. The second term penalizes the agent for leaving the valid area $\mathcal{V}$, which corresponds to a bounded rectangular arena.

**Sim-to-sim gap.** In the previous sim-to-sim environments, we model the sim-to-sim gap by introducing an auxiliary dynamics parameter (e.g., pendulum length) that is not observed during training. In contrast, in the RaceCar environment, the car dynamics in the training environments are governed by a semi-kinematic bicycle model that does not account for interactions between the tire and the ground. On the other hand, in evaluation, we use the dynamical bicycle model of Kabzan et al. (2020). We refer the reader to Kabzan et al. (2020) for the detailed equations of motion as well as to our open-source implementation for more details.

## M  Additional Details on our CMDP Solvers

**CRPO.** CRPO (Xu et al., 2021) is a CMDP solver that uses a primal subgradient switching method to solve the constraint optimization problem. By doing so, CRPO is not tied to any specific RL algorithm and can be used with different policy search methods. For instance, Xu et al. (2021) use this technique with TRPO (Schulman et al., 2015) and Queeney and Benosman (2024) use it with MPO Abdolmaleki et al. (2018). In our experiments, we mainly use CRPO together with SAC (Haarnoja

et al., 2019). To apply CRPO with SAC, we learn a cost critic $Q_c(s, a)$, trained similarly to the reward critic but without the standard entropy term of SAC. To update the policy, we estimate the constraint with $\widehat{C}$, and if it exceeds the budget $d$, we switch from a reward-maximizing update to a cost-minimizing one. The procedure is detailed in Algorithm 2.

---

**Algorithm 2** CRPO with SAC

---

1: **Input:** Constraint budget $d$, replay buffer $\mathcal{D}$
2: **Initialize:** Policy $\pi_\theta$, reward critic $Q_r$, cost critic $Q_c$
3: **for** each environment interaction step **do**
4:     Collect action $a \sim \pi_\theta(\cdot \mid s)$
5:     Observe reward $r$, cost $c$, next state $s'$
6:     Store transition $(s, a, r, c, s')$ in $\mathcal{D}$
7: **end for**
8: **for** each gradient update step **do**
9:     Sample batch $\{(s_i, a_i, r_i, c_i, s'_i)\}_{i=1}^N$ from $\mathcal{D}$
10:     Update reward critic $Q_r$ using SAC critic loss
11:     Update cost critic $Q_c$ using TD error (no entropy bonus)
12:     Estimate constraint $\hat{C} \leftarrow \frac{1}{N} \sum_{i=1}^N Q_c(s_i, \pi_\theta(s_i))$          $\triangleright$ Draw an action from $\pi_\theta$
13:     **if** $\widehat{C} > d$ **then**          $\triangleright$ Constraint violated: prioritize cost minimization
14:         Update policy $\pi_\theta$ using gradient of $\widehat{C}$
15:     **else**          $\triangleright$ Constraint satisfied: optimize for reward
16:         Update policy $\pi_\theta$ using SAC policy loss
17:     **end if**
18: **end for**
19: **return** Policy $\pi_\theta$

---

**Primal-dual solvers.** The primal-dual approach solves CMDPs by augmenting the standard loss used in PPO or SAC with a penalty term derived from the constraint using Lagrangian duality. Specifically, we define the Lagrangian as

$$\mathcal{L}(\theta, \lambda_{\text{PD}}) = J(\pi_\theta) - \lambda_{\text{PD}}\left(C(\pi_\theta) - d\right),$$

where $\lambda_{\text{PD}} \geq 0$ is the Lagrange multiplier. The goal is to find a saddle point of the Lagrangian by performing gradient descent on the policy parameters $\theta$ (primal variables) while performing gradient ascent on $\lambda_{\text{PD}}$ (dual variable) to enforce the constraint.

When integrating this method with SAC, we learn a cost critic together with the reward critic. When computing the policy loss, we evaluate the policy in a same way on the cost critic as we do in the reward critic to evaluate the constraint. The loss is computed via $\pi_\theta = \mathbb{E}_{s,a}[\log \pi_\theta + Q_r(s, \pi() + \lambda_{\text{TD}} Q_c(s, \pi())]$. The same estimate $Q_c(s, \pi$ is used to update the dual

$$\lambda \leftarrow \left[\lambda + \eta_{\lambda_{\text{PD}}}\left(C(\pi_\theta) - d\right)\right]_+,$$

where $\eta_{\lambda_{\text{PD}}}$ is the dual learning rate.

For methods that rely on policy gradient estimators like PPO, we use a value-based estimate of the constraint via a learned cost value function $V_c(s)$, and compute advantage estimates accordingly. The policy update then follows the same principle, where the objective is penalized by the constraint estimate weighted by $\lambda_{\text{PD}}$, ensuring that constraint violations are actively discouraged during training.

More specific implementation details can be found in our open-source implementation.

# N Additional Related Work

**Safe reinforcement learning.** A prominent line of work addressing safety in RL utilizes constrained Markov decision processes. In CMDPs, agents optimize cumulative rewards while satisfying constraints on cumulative costs (Altman, 1999). To solve CMDPs, two main algorithm categories are used: primal-dual and primal methods. Primal-dual approaches leverage the Lagrangian form, aiming to find a saddle-point of it. This is typically done by iteratively updating policy parameters and dual multipliers to balance rewards and constraints (Chow et al., 2018; Achiam et al., 2017; Yang et al., 2020, 2022). On the other hand, primal methods bypass dual variables entirely, focusing on directly optimizing the primal problem by embedding constraints through gradient combination (Xu et al., 2021; Gu et al., 2024). Other works use the CMDP formulation but propose alternative methods to satisfy the safety constraints. Srinivasan et al. (2020) learn a safety actor-critic together with a behavior policy, Thananjeyan et al. (2021) use a recovery backup policy, Bharadhwaj et al. (2020) propose a conservative safety critic together with online rejection sampling of actions. Further discussions on alternative formulations of safety in RL are provided by García and Fernández (2015); Gu et al. (2022); Brunke et al. (2022). This work differs from standard CMDPs in that it focuses on addressing additional model mismatch within the CMDP framework.

**Model uncertainty in reinforcement learning.** Various works address model uncertainty or model mismatch of the environment (e.g., reward function, dynamics, or task itself) during train- and test-time. This challenge is particularly relevant in sim-to-real transfer, which typically serves as a natural motivation for these works. Within this context, a line of research applies robust optimization to optimize worst-case performance, known as robust RL (Iyengar, 2005; Xu and Mannor, 2012; Wolff et al., 2012; Kaufman and Schaefer, 2013; Tamar et al., 2014; Pinto et al., 2017; Pattanaik et al., 2017; Ho et al., 2018; Tessler et al., 2019; Smirnova et al., 2019; Derman and Mannor, 2020; Ho et al., 2021; Badrinath and Kalathil, 2021; Curi et al., 2021; Tanabe et al., 2022; Goyal and Grand-Clement, 2022; Ding et al., 2023; Wang et al., 2023; Sundhar Ramesh et al., 2024). Robust RL algorithms often employ a minimax formulation, significantly increasing design and implementation complexity. Similar to our approach, Gadot et al. (2024) propose an algorithm that avoids explicitly solving the minimax problem, instead relying entirely on how trajectories are sampled. Their key result demonstrates that by sampling next states pessimistically with respect to a reward value function, one can derive a robust policy under a rectangular KL-divergence uncertainty set. In addition, Thomas et al. (2021); As et al. (2022); Zanger et al. (2021) use ensembles and pessimism to enforce safety constraints under model uncertainty as done in SPiDR. This work differs in that our models are not learned from data but are derived via domain randomization. In addition, we formally show that by only using domain randomization, one can achieve safe transfer to the real system. Finally, similar to this work, Yu et al. (2020) propose to penalize rewards based on model uncertainty. However, their focus is in unconstrained offline RL problems. Model uncertainty is also addressed in other formulations such as risk-sensitive RL (Zhang et al., 2023; Kim et al., 2023), domain adaptation in RL (Chen et al., 2024), and curriculum RL (Narvekar et al., 2020). Although these do not focus directly on safety, our work can potentially be adapted to these settings.

**Domain randomization.** Domain randomization enhances policy robustness by training across a variety of environmental scenarios and optimizing average performance. It is widely used in robotics (Sadeghi and Levine, 2016; Tobin et al., 2017; Peng et al., 2018; Andrychowicz et al., 2020). This work utilizes domain randomization as a key component for robustness, integrating it into the constrained RL framework by adding a robust penalty term to the cost function. Given the limited theoretical analysis on domain randomization (Chen et al., 2022), we theoretically study it when facing with additional safety requirements, highlighting its limitations and addressing them. Further, similar to our work, Lee et al. (2023) and Kim et al. (2024) combine CMDPs with domain randomization. However, these works apply CMDPs primarily to legged locomotion, using constraints to shape stylistic gait qualities, rather than ensuring safety. Additionally, their evaluations are largely qualitative and assess constraint satisfaction in simulation. Lastly, a comprehensive survey on domain randomization, and other methods for sim-to-real transfer is discussed in more details by Zhao et al. (2020).

