# OpenReview forum: "SPiDR: A Simple Approach for Zero-Shot Safety in Sim-to-Real Transfer"
_NeurIPS.cc/2025/Conference — NeurIPS 2025 poster_

### Official Review · Reviewer_qaCC · 2025-06-14

**Clarity:** 3
**Significance:** 2
**Originality:** 3
**Rating:** 4
**Confidence:** 4

**Summary:**

This paper studies the problem of ensuring safety in sim-to-real transfer. Existing robust safe reinforcement learning (RL) methods are difficult to be fitted in large-scale training pipelines, while domain randomization technique often cannot ensure safety in practice. This paper proposes an enhanced version of domain randomization called Sim-to-real via Pessimistic Domain Randomization (SPiDR), which approximates the upper bound of the constraint function under the true dynamics by a set of dynamics generated through domain randomization. It is proved that if the upper bounded constraint is satisfied, the constraint under the true dynamics is also satisfied. SPiDR is tested on two real-world robotic tasks and several simulated tasks from mainstream safe RL benchmarks. Results show that SPiDR remains safe when transferred from simulation to the real world without sacrificing too much performance.

**Questions:**

1. How are the next states for variance estimation computed (Line 8 in Algorithm 1)? How the dynamics model be called at any state-action pair?
2. Theorem 4.2 does not consider the approximation of model discrepancy with the variance of next-state predictions. Does the safety guarantee still hold if considering this approximation?
3. In Figure 5, it is wired that the costs of all baselines keep increasing in RaceCar. Is there any explanation for this result?
4. In Experiment 4, how is the optimal performance computed?
5. From Figure 7, it's hard to tell how the time complexity of SPiDR grows with the ensemble size. It's better to first provide a theoretical analysis on time complexity and then compare it with the experiment results. Also, analysis on memory complexity is needed.

**Ethical Concerns:**

["NO or VERY MINOR ethics concerns only"]

**Final Justification:**

The authors responded to four of my five questions. Questions 2, 3, and 4 are addressed. The response to Question 1 indicates that the algorithm does require the simulator to support next-state computation at an arbitrary state, which I believe is a limitation.

**Limitations:**

yes

**Quality:**

3

**Strengths And Weaknesses:**

Strengths:

1. Theoretical guarantee on the safety during sim-to-real transfer is provided.
2. The proposed algorithm is validated in real-world robotic tasks, showing potential real-world applications.
3. This paper is written clearly. The methods, theories, and experiments are well described and easily understandable.

Weaknesses:

1. The proposed algorithm requires the dynamics model to be directly callable for estimating the next-state prediction variance. That is, given an arbitrary state-action pair, the simulator can return a next state. This requirement cannot be satisfied by many existing simulators, which only supports rollout from a randomly chosen initial state.
2. The safety guarantee provided by Theorem 4.2 does not consider the approximation of model discrepancy with the variance of next-state predictions introduced in Section 4.2.

---

> ### Author Rebuttal · Authors · 2025-07-30
>
> We would like to thank you for carefully reviewing our paper and for your valuable feedback! Please find below our response to some of your questions.
>
> > The proposed algorithm requires the dynamics model to be directly callable for estimating the next-state prediction variance.  That is, given an arbitrary state-action pair, the simulator can return a next state. This requirement cannot be satisfied by many existing simulators, which only supports rollout from a randomly chosen initial state.
>
> Thank you for this important question. To the best of our knowledge, state-of-the-art simulators support setting the system's state arbitrarily out of the box:
> 1. In IsaacGym [1] via the `set_actor_root_state_tensor` API.
> 2. In DeepMind Control Suite [2] (and MuJoCo) in example:
> ```
>     with physics.reset_context():
>         # Set cart and pole initial state [cart_position, cart_velocity, pole_angle, pole_angular_velocity]
>         physics.named.data.qpos["slider"] = init_state[0]  # Cart position
>         physics.named.data.qvel["slider"] = init_state[1]  # Cart velocity
>         physics.named.data.qpos["hinge_1"] = init_state[2]  # Pole angle
>         physics.named.data.qvel["hinge_1"] = init_state[3]  # Pole angular velocity
>     env._task.after_step(physics)
> ```
> 3. In Brax [3] and Mujoco Playground [4] environments by just calling step, as environments are purely-functional (i.e. stateless), which allows one to directly call the step function with any state-action pair.
>
> While our implementation uses Brax and MujocoPlayground, using other simulators is possible. We are keen to provide an implementation of SPiDR in other simulators such as IsaacGym, if there is interest from the robotics and safe RL communities.
>
> > Theorem 4.2 does not consider the approximation of model discrepancy with the variance of next-state predictions. Does the safety guarantee still hold if considering this approximation?
>
> In appendix C we formally analyze the conditions under which our empirical estimator provides a guarantee for Theorem 4.2. Please let us know if we should highlight this point further in our main text—we are happy to do so.
>
> > In Figure 5, it is wired that the costs of all baselines keep increasing in RaceCar. Is there any explanation for this result?
>
> Thank you for raising this interesting question. We note that the Nominal baseline fails simply because it is trained only on the nominal environment, which has different dynamics from the training environment. Domain Randomization fails due to underestimation in the constraint, as shown in Lemma 4.1. Both of these baselines are expected to fail, which leaves us only with RAMU. We believe that the reason for RAMU's failure on this task, relates to how it computes the next-state prediction when updating its risk-averse Bellman update (see [1], page 7). In short, these next-states are sampled as a (random) convex combination between the current state and the next state that was originally predicted by the simulator. In the RaceCar environment, the training dynamics significantly deviate from the evaluation dynamics since they only use a \emph{kinematic} model, not accounting for tire and motor dynamics. We hypothesize that by taking a convex combination of the current state and the next one, RAMU cannot capture possible states that lie outside the line connecting the current and next state, therefore possibly failing to penalize states that are not seen during training. SPiDR on the other hand uses domain randomization, therefore has possibly richer set of states to learn from.
>
> > In Experiment 4, how is the optimal performance computed?
>
> We compute the optimal performance by running SAC with CRPO or a primal-dual method (see also Appendix M) on the test dynamics. To the best of our knowledge, these results match the established benchmark performance on these tasks [4]. We will clarify this point in our revised version. Thank you for this question.
>
> We hope our response addresses all of your concerns, and would appreciate it if you would consider increasing our score. We are happy to answer any other open questions. Thanks again for your active engagement!
>
> ---
>
> [1] Makoviychuk, Viktor, Lukasz Wawrzyniak, Yunrong Guo, Michelle Lu, Kier Storey, Miles Macklin, David Hoeller et al. "Isaac gym: High performance gpu-based physics simulation for robot learning." arXiv preprint arXiv:2108.10470 (2021).
>
> [2] Tassa, Yuval, Yotam Doron, Alistair Muldal, Tom Erez, Yazhe Li, Diego de Las Casas, David Budden et al. "Deepmind control suite." arXiv preprint arXiv:1801.00690 (2018).
>
> [3] Freeman, C. Daniel, Erik Frey, Anton Raichuk, Sertan Girgin, Igor Mordatch, and Olivier Bachem. "Brax--a differentiable physics engine for large scale rigid body simulation." arXiv preprint arXiv:2106.13281 (2021).
>
> [4] Zakka, Kevin, Baruch Tabanpour, Qiayuan Liao, Mustafa Haiderbhai, Samuel Holt, Jing Yuan Luo, Arthur Allshire et al. "Mujoco playground." arXiv preprint arXiv:2502.08844 (2025).
>
> [5] Queeney, James, and Mouhacine Benosman. "Risk-averse model uncertainty for distributionally robust safe reinforcement learning." Advances in Neural Information Processing Systems 36 (2023): 1659-1680.

---

> > ### Comment · Reviewer_qaCC · 2025-08-03
> > **Comment for rebuttal**
> >
> > Thanks for your response. Most of my questions are addressed, and I will keep my score.

---

> ### Author Response · Authors · 2025-08-03
>
> Thank you for your follow-up and for acknowledging that most of your concerns have been addressed.
>
> If there are any remaining points you'd like us to further clarify or elaborate on, we'd be very keen to do so and are open for additional suggestions and feedback to help us improve our paper.
>
> Thank you again for the time and effort you've put into reviewing out paper.

---

### Official Review · Reviewer_5h5Y · 2025-06-26

**Clarity:** 3
**Significance:** 2
**Originality:** 3
**Rating:** 4
**Confidence:** 4

**Summary:**

This paper addresses the critical challenge of zero-shot safe sim-to-real transfer in reinforcement learning, where a policy trained entirely in simulation must satisfy safety constraints upon its first deployment in the real world. The proposed SPIDR builds on domain randomization by incorporating a pessimistic penalty into the safety constraint. Specifically, SPIDR approximates this uncertainty by measuring the variance of next-state predictions from an ensemble of simulators and adds this variance as a penalty to the cost function during training. This encourages the agent to avoid state-action regions where the simulator is unreliable, leading to more robustly safe policies.

**Questions:**

1) The paper's framework assumes that the dynamics $p$ differ between sim and real, does the difference of cost function $c$ is also considered? If not, the "cost-to-real" gap can also exist due to sensor noise, calibration errors, or differing physics of cost-inducing events (e.g., contacts). How robust do you expect SPIDR to be if the cost function itself is also mismatched?
2) The success of SPIDR hinges on the pessimism parameter λ, and the tuning guide in Appendix E suggests an iterative process on the real system. This seems to conflict with the paper's "zero-shot" premise. Could you please elaborate on this process? Specifically, for the RaceCar and Unitree Go1 experiments, how many real-world rollouts or trials were required to select the final λ values reported in the paper?
3) SPIDR increases the cost, which can potentially render the CMDP infeasible if λ is too high or the sim-to-real gap is too large. This could lead to the agent learning a trivial "freezing" policy to avoid all costs. Take an example that might not be very accurate: what if the $\lambda v(s_t, a_t) > d$? Did you encounter this failure mode in your experiments?

**Ethical Concerns:**

["NO or VERY MINOR ethics concerns only"]

**Final Justification:**

The authors provided a good rebuttal that addressed most of my initial concerns.

* Concern on "Zero-Shot": My primary concern about the "zero-shot" claim was resolved. The authors' clarification that it refers to safety on first deployment (not optimal performance) is a crucial distinction.  This, combined with their confirmation that the initial real-world deployments were indeed safe without prior tuning, validates their central claim.

* Concern on Scalability: The authors conducted a new vision-based experiment that demonstrates the method's applicability, turning a potential weakness into a strength.

Since authors has provided extensive experiment and addressed most of my concerns, and I will raise my score.

**Limitations:**

Yes.

**Paper Formatting Concerns:**

No.

**Quality:**

3

**Strengths And Weaknesses:**

Strengths:
1) The paper is very well-written, structured, and easy to understand. The motivation is laid out clearly, and the progression from the theoretical concept to the final algorithm design is logical and intuitive .
2) The proposed method is well-founded in theory. It provides a formal upper bound on the real-world cost (Lemma 4.1) and a clear safety guarantee (Theorem 4.2) showing that policies trained with SPIDR will satisfy constraints in the real world.
3) The deployment in real-world scenarios validates its ability of zero-shot transfer of sim2real.

Weaknesses:
1) The primary weakness lies in the practical requirement to tune the pessimism hyperparameter λ. The paper provides a heuristic for selecting λ in Appendix E, but this process involves iterative testing on the physical system to find a suitable balance between safety and performance. This need for real-world tuning somewhat contradicts the "zero-shot" claim and could be costly or impractical in many real-world scenarios.
2) The effectiveness of the empirical variance as an estimator depends on the ensemble size n. For systems with very high-dimensional state spaces (e.g., from vision), a much larger ensemble might be required to get a reliable uncertainty estimate, which could impact computational performance. The paper's experiments are limited to robotics tasks with proprioceptive states, and it does not analyze how the method would scale to such high-dimensional problems.

---

> ### Author Rebuttal · Authors · 2025-07-30
>
> Thank you for thoroughly reviewing our paper and for helping us improve it.
> Please find below our detailed responses to the concerns raised.
>
> > The proposed method is well-founded in theory. It provides a formal upper bound on the real-world cost (Lemma 4.1) and a clear safety guarantee (Theorem 4.2) showing that policies trained with SPIDR will satisfy constraints in the real world.
> The deployment in real-world scenarios validates its ability of zero-shot transfer of sim2real.
>
> Thank you acknowledging these aspects of our work!
>
> > The primary weakness lies in the practical requirement to tune the pessimism hyperparameter $\lambda$. The paper provides a heuristic for selecting $\lambda$ in Appendix E, but this process involves iterative testing on the physical system to find a suitable balance between safety and performance.
>
> We recognize that selecting an appropriate value for $\lambda$ is critical to the success of SPiDR. Notably, incorporating domain knowledge, such as insights into simulator fidelity or the CMDP's cost function makes the selection process more straightforward in practice. Additionally, tuning $\lambda$ was usually not the main bottleneck in tuning parameters, but rather first finding policies that satisfy the constraints in simulation, e.g., tuning primal-dual parameters, or parameters that relate to SAC/PPO.
>
> In addition, we highlight that the goal of the paper is satisfy the constraints on the real system upon deployment.
> More formally, we give a safety guarantee and not a performance guarantee for SPiDR. In Appendix E we show how after the first safe deployment, one can indeed improve performance by gradually reducing $\lambda$. This appendix should rather be viewed as an additional guidance for practitioners when using SPiDR, instead of a formal statement about SPiDR's safety guarantees. We apologize for this confusion and wil clarify this ambiguity in the main text of our revised version. We will additionally clarify in our revised version that the term “zero-shot” refers only to safety, and not to performance. Thank you for this comment and for helping us improve the presentation of our work!
>
> > For systems with very high-dimensional state spaces (e.g., from vision), a much larger ensemble might be required to get a reliable uncertainty estimate, which could impact computational performance.
>
> Figure 7 indicates a “diminishing returns” trend when increasing $n$ beyond $8$, while generally requiring feasible computation resources, even on the Humanoid task with observations of 244 dimensions. Since training happens in simulation, where privileged state information is available, one can still penalize the cost w.r.t. variance in states, instead of penalizing disagreement of images, even if the policy operates on these images without direct access to the states. Such asymmetric training, via a teacher-student architecture, is quite common in practice [1, 2].
> We agree that the addition of vision control experiments would improve our work. Following your suggestion, we conducted an additional experiment, implementing the asymmetric architecture described above. We use DrQ [3] with CRPO [4] as a penalizer and run it on the CartpoleSwingup task. The policy operates on a sequence of 3 stacked greyscale images, while the cost is penalized with SPiDR by using the privileged system states that are not shown to the agent but are known in simulation. We provide our results below.
> |           | Cumulative Cost (± SE)    | Cumulative Reward (± SE)  |
> |-----------|---------------------------|----------------------------|
> | Domain Randomization | 267.156 ± 405.2127        | 872.506 ± 64.9             |
> | SPiDR     | 44.8047 ± 20.8520         | 674.608 ± 85.54            |
>
> As shown, even in a partially observable setup, where the policy operates on a 64x64x3 image, penalizing the cost with privileged data (available in simulation), allows the agent to satisfy the constraint, even under a significant model mismatch.
> Training a vision-based policy for 1M time steps with SPiDR, using an ensemble of $n = 8$ environments, takes roughly 33 minutes on an RTX 4090 GPU. Training the same setup without SPiDR takes about 30 minutes and the main bottleneck is in computing the critic's gradients. This speedup is achieved by using Madrona-MJX, allowing us to render rollouts from 128+ environments in parallel. Thank you for this great question! We believe the addition of this experiment gives SPiDR an interesting direction which we did not think about originally.
>
> > The paper's framework assumes that the dynamics differ between sim and real, does the difference of cost function is also considered?
>
> We mainly consider perturbations in the dynamics (referred to as “domain randomization”), as it seems, at least empirically, that such perturbations play a critical role in successful sim-to-real transfer across many real-world tasks. Theoretically speaking, our framework can naturally be generalized for mismatches in the cost function. Assuming bounded mismatch between the simulated and the real cost functions, one can show that an additional term relating to this mismatch will appear in Equation 4. Intuitively, as this gap increases, the feasible set of policies shrinks, making it harder to solve the task. However, in practice, such modeling discrepancies are typically modest, allowing us to to compute safe policies in simulation.
> Thank you for raising this question, this is an interesting aspect we did not consider, and we will add a discussion on that in the appendix.
>
> > Specifically, for the RaceCar and Unitree Go1 experiments, how many real-world rollouts or trials were required to select the final $\lambda$ values reported in the paper?
>
> For both tasks our initial choice of $\lambda$ (i.e., the largest one) satisfied the safety constraints, as shown in Figure 14. This is in line with our theory and the main claim of the paper. Our initial choices of $\lambda$ are based on our domain knowledge about the sim-to-real gap of the problems. For instance, the lower value of $\lambda$ for the Go1 was indeed chosen since the simulator we have for this task is much better compared to the one of the RaceCar (mainly due to difficulties in modeling tire dynamics).
>
> > SPIDR increases the cost, which can potentially render the CMDP infeasible if $\lambda$ is too high or the sim-to-real gap is too large. Did you encounter this failure mode in your experiments?
>
> We appreciate this valuable comment. Theoretically speaking, it is possible to think of problems where the sim-to-real gap is large enough so that the feasible set of the penalized CMDP (Equation 5) is empty. More generally, we believe that many real-world scenarios are a bit more optimistic than the theoretical worst-case: simulators are becoming better in many tasks and therefore an optimal policy within the feasible set of the penalized CMDP might not deviate too much from an optimal policy of the real CMDP. Empirically, in our sim-to-real experiments, for increasing values of $\lambda$, policies qualitatively seemed to be more “cautions”. In example, in the RaceCar task, increasing $\lambda$ led to policies that drove slower, causing the car to sometimes fail to reach the goal within the specified time horizon. This result is captured quantitatively by the lower costs and rewards in Figure 14.
>
> We believe our response above answers your questions and would appreciate it if you would consider increasing your score. Please let us know if you have any further questions, we would be very happy to answer.
>
> ---
>
> [1] Chen, Fu, Rui Wan, Peidong Liu, Nanxing Zheng, and Bo Zhou. "VMTS: Vision-Assisted Teacher-Student Reinforcement Learning for Multi-Terrain Locomotion in Bipedal Robots." arXiv preprint arXiv:2503.07049 (2025).
>
> [2] Lee, Joonho, Jemin Hwangbo, Lorenz Wellhausen, Vladlen Koltun, and Marco Hutter. "Learning quadrupedal locomotion over challenging terrain." Science robotics 5, no. 47 (2020): eabc5986.
>
> [3] Yarats, Denis, Ilya Kostrikov, and Rob Fergus. "Image augmentation is all you need: Regularizing deep reinforcement learning from pixels." In International conference on learning representations. 2021.
>
> [4] Xu, Tengyu, Yingbin Liang, and Guanghui Lan. "Crpo: A new approach for safe reinforcement learning with convergence guarantee." In International Conference on Machine Learning, pp. 11480-11491. PMLR, 2021.

---

> > ### Author Response · Authors · 2025-08-05
> >
> > Dear reviewer 5h5Y
> >
> > Thank you again for the comments on our paper.
> >
> > We would like to ask whether our response addressed your concerns and to offer further clarification if requested. We would be very pleased for an open discussion regarding the points brought up. Please reach out if you have any remaining concerns, as the discussion period ends soon.

---

> > ### Comment · Reviewer_5h5Y · 2025-08-05
> >
> > Thank you for your rebuttal. I believe most of my concerns are addressed.
> > I was impressed by the new vision-based experiment demonstrating the method's scalability.
> > I will raise my score to 4.

---

> > > ### Author Response · Authors · 2025-08-05
> > >
> > > We appreciate your thoughtful feedback and are glad we were able to address your concerns. Thank you as well for raising your score.
> > >
> > > We genuinely believe that the vision-based experiment is a very important addition to our paper. This aspect was not initially considered during the writing process, and we are grateful to you for bringing it to our attention.

---

### Official Review · Reviewer_HvGa · 2025-06-30

**Clarity:** 3
**Significance:** 3
**Originality:** 2
**Rating:** 4
**Confidence:** 4

**Summary:**

This paper introduces SPIDR, a novel algorithm designed to address the challenge of zero-shot safe sim-to-real transfer in reinforcement learning. The authors state that standard domain randomization is scalable but typically unsafe, while formal robust methods tend to be too complex for practical use and scale. SPIDR addresses this gap by augmenting standard domain randomization with a pessimistic penalty. The idea behind this pessimism term is that uncertain transitions, i.e., transitions that result in a variety of successor states under an ensemble of randomized simulators, should be penalized so as to avoid exploitation of uncertain dynamics. By adding this state-dependent uncertainty measure as a negative bonus to the safety cost function, SPIDR encourages the learned policy to avoid regions where a variety of simulators do not result in the same successor state, thereby acting more conservateively and improving safety upon real-world deployment. The method is evaluated empirically on a variety of simulated benchmarks and, notably, on two distinct real-world robotic platforms.

**Questions:**

My main questions relate to the weaknesses I have outline above:

- Could you clarify your definition of "zero-shot" in the context of this paper? And does it include the tuning process of $\lambda$? A more explicit discussion of this definition in the main text would be valuable.
- How would you expect SPIDR to perform against a simple baseline as the one I outlined above (i.e., vanilla DR where the cost budget itself is treated as a tunable hyperparameter that is allowed to be tuned online)? A discussion of state-dependent vs. uniform pessimism could also be insightful in general.
- How would you expect SPIDR to perform in highly stochastic environments and are there ways to circumvent potential issues? See above for my reasoning.

I am willing to raise my score if the mentioned points are addressed clearly in a revision.

**Ethical Concerns:**

["NO or VERY MINOR ethics concerns only"]

**Final Justification:**

The authors have provided an extensive rebuttal, including additional empirical studies, that addressed many of my initial concerns. Specifically, the authors addressed and clarified
- concerns about the online tuning of the pessimism parameters and the implication on the nature of the "zero-short" property
- differentiation to related works
- additional empirical ablations / experiments to an additional intuitive baseline

I believe the online tuning of pessimism still requires future investigation, but this is to be expected and would not be a reasonable requirement for this submission. I do believe the additional of models that could seperate aleatoric and epistemic uncertainty to deal with stochastic systems more fundamentally would be a sensible, method-specific addition to SPIDR. Overall, I find that enough questions / weaknesses have been addressed to raise my score. I thank the authors for their work and the extensive rebuttal.

**Limitations:**

Yes. However, a few points, e.g., the need for real-world trials to tune $\lambda$ or stochastic transitions (if I understood correctly) may be addressed more centrally.

**Paper Formatting Concerns:**

No concerns.

**Quality:**

3

**Strengths And Weaknesses:**

Strengths:

- Significance: The paper addresses a problem of high importance to the field, the reliable deployment of simulated agents to real-world applications. Ensuring safety in sim-to-real transfer is a bottleneck for deploying RL in real-world applications, and a practical, scalable solution may be impactful for both researchers and practitioners.

- Clarity: This paper is written clearly, and the approach is simple conceptually, indicating that it may be easy to integrate into existing pipelines. It modifies standard domain randomization setups in a relatively minimal way, which is by altering the cost signal. Relevant background is explained well and the derivations are straightforward to follow.

- Quality: The empirical evaluation of the proposed evidence is strong, in particular due to the inclusion of real-world experiments with robotic sim-to-real settings. The experiments support the paper's claim, albeit more baseline methods or ablation experiments (see below) could have been run. I believe, however, that this is acceptable given that they are real-world experiments.  is exceptionally strong.
- The theoretical grounding looks sound in principle. The practical algorithm however deviates from the theoretical grounding in a number of ways (see below). I believe fairly similar related bounds (albeit not exactly the same) have been proposed in the literature I believe.

Weaknesses:

- Originality: The algorithmic novelty seems somewhat limited to me. While the specific application to the model-free sim-to-real setting may be novel, the concept of using ensemble variance as an uncertainty penalty to ensure safety is a rather well-established pattern for example in the literature on safe model-based RL. I believe this line of research is discussed rather sparingly in the current form of the paper, for example several works derive highly related bounds and algorithmic components to the proposed ones that precede the cited work by Sun et al.:
- Safe continuous control with constrained model-based policy optimization (Zanger et al., 2021)
- Constrained Policy Optimization via Bayesian World Models (As et al. 2022)
- Safe Reinforcement Learning by Imagining the Near Future (Thomas et al. 2021)
the paper would be strengthened by discussing and differentiating itself from these works in a bit more detail.

- Parameter tuning: An algorithmic weakness in my view lies in the practical need to tune the pessimism parameter $\lambda$. Parameters of this kind are generally known to be hard to tune, which warrants the use of Lagrangian methods and constrained solvers with access to online data many works in the literature. It is commendable that the authors suggest a procedure for finding this parameter (Appendix E), but from what I understand, this procedure requires tests on the online MDP. In my view, this procedure also contradicts the definition of "zero-shot" transfer since it seems likely that one requires at least a "few shots" to find an appropriate pessimism level. This distinction should be made more explicit, or (even better) be addressed by directly the inference of $\lambda$ as an algorithmic component (e.g., by phrasing the problem as a POMDP). I do, however, realize that the latter would be a rather tall ask for a rebuttal.

- Experimental baseline: The above point also affects the experimental evaluation. The comparison with the standard domain randomization is important but feels slightly imbalanced as is due to the presence of the additional tunable hyperparameter. An extremely simple baseline that would be granted a similar tuning pipeline for a pessimism parameter $\lambda$ could use vanilla domain randomization but simply changes the admissable cost limit as a hyperparameter.

- Stochastic environments: It seems that the current practical implementation of the algorithm would be very susceptible to stochastic transitions, since it replaces the wasserstein distance $D_W$ with variances. For highly stochastic environments, this should result in overly conservative agents if I am not wrong.

---

> ### Author Rebuttal · Authors · 2025-07-30
>
> We would like to thank you for carefully reviewing our paper and for your clear and constructive feedback! You can find below our response to your questions.
>
> > Quality: The empirical evaluation of the proposed evidence is strong, in particular due to the inclusion of real-world experiments with robotic sim-to-real settings. is exceptionally strong. Significance: The paper addresses a problem of high importance to the field, the reliable deployment of simulated agents to real-world applications.
>
> Thank you again for acknowledging our work and for the recognition of our contributions.
>
> > Could you clarify your definition of "zero-shot" in the context of this paper? And does it include the tuning process of $\lambda$? A more explicit discussion of this definition in the main text would be valuable.
>
> Thank you for pointing this out. To be concrete, our definition of “zero-shot” relates only to _safety_. More specifically, under standard regularity assumptions, one can guarantee safe transfer on the first trial. We do not state however guarantees about the _performance_ upon transfer, precisely because attaining optimal performance may require additional online trials on the real system.
>
> In appendix E, we provide an informal procedure that serves as a recommendation to practitioners. In this procedure, safe transfer is guaranteed on the first trial, while performance can be improved after additional post-simulation trials. We agree that this manual process should be automated, e.g. by running online safe Bayesian Optimization techniques [1, 2, 3]. A similar approach has been recently proposed in [4], in the context of safe offline learning and is highly relevant, yet complementary to SPiDR. We view an online procedure for the selection of $\lambda$ as a highly valuable direction for future work.
>
> In a broader sense, the choice of hyperparameters in the context of safe learning is a critical, yet an extremely challenging problem, which we do not aim to tackle. For example, in SafeOpt [1, 2], a well-established algorithm with probabilistic guarantees for constraint satisfaction during learning, the choice of GP kernel and Lipschitz constants are left to the practitioner. In a similar vain, $\lambda$ is in fact derived from the Lipschitz constant of the constraint, and can be viewed intuitively as a prior over the sim-to-real gap. To the best of our knowledge, an initial attempt to address safe hyperparameters selection was done in [5, FrontierSearch], yet significant work still remains to be done. Based on your suggestion and the discussion above, we will clarify further in the main text of the paper, that zero-shot safety relates only to ensuring safety on the first trial, without guaranteeing optimality on the real system.
>
> > How would you expect SPIDR to perform against a simple baseline as the one I outlined above? A discussion of state-dependent vs. uniform pessimism could also be insightful in general.
>
> Thank you for raising this important question.
> There are two main challenges with tuning the budget (“constraint tightening”). First, severe tightening may yield a “zero budget” training regime that CMDP solvers like CRPO and primal-dual methods struggle to solve in practice [6, 7]. Second, as hinted, reducing the budget is equivalent to finding a uniform upper-bound to the penalty term in Equation 4, that does not depend on state-actions (“uniform pessimism”). Therefore, in order to achieve the same level of penalty required to satisfy the constraint, uniform pessimism can hurt performance by being over-pessimistic in those states where the uncertainty about the sim-to-real gap might in fact be low, effectively penalizing the “wrong” states. We visualize such state-action-dependent uncertainty in Figure 13.
> Thank you for the great suggestion to include this discussion in the paper, we will add it to Appendix D.
>
> Following your suggestion, we have also conducted an additional ablation on the real-world race car system, repeating our experiment with five trials per seed across five seeds, testing constraint tightening with values $d \in \{0, 7.5\}$ (where the standard budget is $d = 15$). We present below the mean and standard error across the five seeds (on the real system)
>
> |  | Cumulative Cost (± SE)    | Cumulative Reward (± SE)  |
> |----------|---------------------------|----------------------------|
> | d = 0        | 58.7074 ± 10.5819         | 14.1819 ± 3.3733           |
> | d = 7.5      | 141.1890 ± 29.7231        | 29.1217 ± 5.9963           |
> | SPiDR     | 10.7956 ± 1.2856          | 62.6296 ± 6.5359           |
>
> As shown, in both settings, SPiDR outperforms constraint tightening both in terms of safety and performance. We note that for $d =0$, the constraint was not satisfied even in simulation, precisely as described in [6].
>
> In total, we compare SPiDR _on hardware_ with three other baselines: RAMU (Appendix G), Constraint Tightening (will be added in Appendix H) and Domain Randomization (main text). This additional experiment strengthens our confidence in SPiDR's effectiveness in satisfying constraints upon dynamics mismatch, supporting our main contribution: providing a simple, yet effective algorithm for safe sim-to-real transfer.
>
> > How would you expect SPIDR to perform in highly stochastic environments and are there ways to circumvent potential issues? See above for my reasoning.
>
> Indeed, our implementation uses only one sample per dynamics model, which could add conservatism in highly stochastic environments. In principle, one would need to separate the aleatoric noise from uncertainty that originates from model ambiguity. This can be done by drawing several samples from each model and marginalizing the noise [8]. We opt for using a single sample as simplicity is an important design choice of SPiDR. Empirically, we observe that even on very stochastic environments, such as the Go1 (noise added to observations, commands are sampled randomly) and PointGoal2 (target positions are resampled during rollouts), SPiDR works well, outperforming other baselines and transferring safely to a real Go1 robot while maintaining competitive performance.
>
> > several works derive highly related bounds and algorithmic components...the paper would be strengthened by discussing and differentiating itself from these works in a bit more detail.
>
> We agree with this statement; our theoretical argument relies at its core on the simulation lemma, which is indeed a standard tool for analyzing many other related problems. We will add a paragraph in the related works section discussing pessimism in the context of model-based CMDP solvers, differentiating more precisely our contribution relating to the important works mentioned above. Thank you for bringing this to our attention and helping us improve our paper!
>
> We thank you once again for your feedback and hope our responses adequately address your concerns. We'd be happy if you would consider revising our score, and we are happy to answer any other open questions. Thanks again for your feedback and taking the time to help us improve our work.
>
> ---
>
> [1] Berkenkamp, Felix, Andreas Krause, and Angela P. Schoellig. “Bayesian optimization with safety constraints: safe and automatic parameter tuning in robotics.” Machine learning 112, no. 10 (2023): 3713-3747.
>
> [2] Sui, Yanan, Alkis Gotovos, Joel Burdick, and Andreas Krause. “Safe exploration for optimization with Gaussian processes.” In International conference on machine learning, pp. 997-1005. PMLR, 2015.
>
> [3] Kaushik, Rituraj, Karol Arndt, and Ville Kyrki. “Safeapt: Safe simulation-to-real robot learning using diverse policies learned in simulation.” IEEE Robotics and Automation Letters 7, no. 3 (2022): 6838-6845.
>
> [4] Wachi, Akifumi, Kohei Miyaguchi, Takumi Tanabe, Rei Sato, and Youhei Akimoto. “A Provable Approach for End-to-End Safe Reinforcement Learning.” arXiv preprint arXiv:2505.21852 (2025).
>
> [5] Rothfuss, Jonas, Christopher Koenig, Alisa Rupenyan, and Andreas Krause. “Meta-learning priors for safe Bayesian optimization.” In Conference on robot learning, pp. 237-265. PMLR, 2023.
>
> [6] He, Tairan, Weiye Zhao, and Changliu Liu. “Autocost: Evolving intrinsic cost for zero-violation reinforcement learning.” In Proceedings of the AAAI Conference on Artificial Intelligence, vol. 37, no. 12, pp. 14847-14855. 2023.
>
> [7] Huang, Weidong, Jiaming Ji, Chunhe Xia, Borong Zhang, and Yaodong Yang. “Safedreamer: Safe reinforcement learning with world models.” arXiv preprint arXiv:2307.07176 (2023).
>
> [8] Depeweg, Stefan, Jose-Miguel Hernandez-Lobato, Finale Doshi-Velez, and Steffen Udluft. "Decomposition of uncertainty in Bayesian deep learning for efficient and risk-sensitive learning." In International conference on machine learning, pp. 1184-1193. PMLR, 2018.

---

### Official Review · Reviewer_3yuV · 2025-07-03

**Clarity:** 3
**Significance:** 3
**Originality:** 3
**Rating:** 5
**Confidence:** 2

**Summary:**

This paper presents SPiDR, a method for safe zero-shot sim-to-real transfer in reinforcement learning. Motivated by the safety risks from the sim-to-real gap, SPiDR combines domain randomization with a pessimistic cost penalty based on uncertainty from an ensemble of simulators. This penalization ensures policies avoid risky regions during training. Experiments on both simulated benchmarks and real robots (a race car and a quadruped) show that SPiDR achieves safe and effective transfer, outperforming standard domain randomization and robust RL baselines.

**Questions:**

How sensitive is SPiDR to the choice of the pessimism coefficient (λ), and how practical is it to tune this parameter in real-world applications?
Does SPiDR still ensure safety if the real environment lies significantly outside the range of the domain randomization distribution?
What are the limitations of using ensemble variance as a proxy for model uncertainty? Are there alternative metrics that might work better?

**Ethical Concerns:**

["NO or VERY MINOR ethics concerns only"]

**Final Justification:**

This work demonstrates a sim-to-real transfer method with robust RL with theoretically guarantees.
The rebuttal has addressed my concerns over how to tune the lambda parameter and scalability. I would like to maintain my original recommendation of accept.

**Limitations:**

yes

**Quality:**

3

**Strengths And Weaknesses:**

Strengths:
- SPiDR provides provable zero-shot safety during sim-to-real transfer without relying on real-world data
- SPiDR is theoretically sound
- It is simple to implement and integrates easily with existing RL methods
- It shows strong performance in both simulated benchmarks and real-world robot tasks

Weaknesses:
- The pessimistic penalty can lead to overly conservative policies, potentially reducing task performance.
- SPiDR requires manual tuning of the pessimism parameter

---

> ### Author Rebuttal · Authors · 2025-07-30
>
> Thank you for your thoughtful review and accurate summary of our paper! We appreciate your acknowledgment that SPiDR “shows strong performance in both simulated benchmarks and real-world robot tasks”. Please find below our response to some of your questions.
>
> > The pessimistic penalty can lead to overly conservative policies, potentially reducing task performance.
>
> We fully agree with this statement. When the sim-to-real gap is not precisely known, we believe safety should be maintained first, and only then performance is ought to be improved by gradually reducing conservatism with online data. SPiDR allows practitioners to achieve the first step in this procedure.
>
> > how practical is it to tune this parameter in real-world applications?
>
> We acknowledge that the choice of $\lambda$ is crucial for the success of SPiDR. In addition, we would like to point out, that since $\lambda$ has a physical interpretation (see Equation 4), domain knowledge about the problem, such as knowledge about the fidelity of the simulator and the cost function of the CMDP, can greatly simplify this process.
> For instance, in both of our real-world experiments, we initially select $\lambda$ by matching the scale of the penalty (in simulation) to be roughly the same scale of the cost function. The simulator of the Go1 robot is much more accurate compared to the RaceCar, and therefore it required lower values for $\lambda$. Thank you for raising this important question! We will clarify this point in the paper by including the discussion above to Appendix E.
>
> > Does SPiDR still ensure safety if the real environment lies significantly outside the range of the domain randomization distribution?
>
> Thank you for raising this question. Yes, this is exactly how we conduct our sim-to-sim experiments. The parameters used to perturb the environment in evaluation are different than those used in domain randomization during training. For instance, in the Walker task, we randomize only the motor gear parameters, while in evaluation we additionally also randomize the torso length. This is mentioned in detail in Appendices J-L, however we will clarify this point further in the main text of our revised version.
>
> We greatly appreciate your feedback and we are happy to address any further questions or comments.

---

> > ### Author Response · Authors · 2025-08-05
> >
> > Dear reviewer 3yuV
> >
> > Thank you again for the comments on our paper.
> >
> > We would like to ask whether our response addressed your concerns and to offer further clarification if requested. We would be very pleased for an open discussion regarding the points brought up.

---

> ### Comment · Reviewer_3yuV · 2025-08-05
> **Thanks for the response**
>
> I have read the authors' response to my concerns and would like to maintain my original assessment.

---

> > ### Author Response · Authors · 2025-08-05
> >
> > Thank you again for taking the time to review our paper and for your positive evaluation of our work.

---

### Decision · Program_Chairs · 2025-09-17

**Decision:**

Accept (poster)

**Comment:**

This paper presents SPiDR, an effective method for safe zero-shot sim-to-real transfer in reinforcement learning. The approach is supported by theoretical guarantees and validated on real-world robotic platforms, demonstrating strong performance. The paper is well-written and easy to follow, and it received unanimous positive reviews. While reviewers raised concerns about parameter tuning (with respect to lambda), simulator querying issue, and scalability to high-dimensional problems, the authors effectively addressed the main issues during the rebuttal and discussion by providing clarifications along with new vision-based and hardware experiments. Overall, this is a good paper that merits acceptance.